# How Explanations Leak the Decision Logic: Stealing Graph Neural Networks via Explanation Alignment

## Abstract

Graph Neural Networks (GNNs) have become essential tools for analyzing graph-structured data in domains such as drug discovery and financial analysis, leading to growing demands for model transparency. Recent advances in explainable GNNs have addressed this need by revealing important subgraphs that influence predictions, but these explanation mechanisms may inadvertently expose models to security risks. This paper investigates how such explanations potentially leak critical decision logic that can be exploited for model stealing. We propose EGSteal, a novel stealing framework that integrates explanation alignment for capturing decision logic with guided data augmentation for efficient training under limited queries, enabling effective replication of both the predictive behavior and underlying reasoning patterns of target models. Experiments on molecular graph datasets demonstrate that our approach shows advantages over conventional methods in model stealing. This work highlights important security considerations for the deployment of explainable GNNs in sensitive domains and suggests the need for protective measures against explanation-based attacks. Our code is available at `https://anonymous.4open.science/r/EGSteal-2BF7`.

## 1 Introduction

Graph Neural Networks (GNNs) have emerged as powerful tools for analyzing graph-structured data, demonstrating remarkable success across various critical scenarios including drug discovery Sun et al. (2020), financial analysis Lv et al. (2019), and social network analysis Hamilton et al. (2017). The need for trust in these high-stakes scenarios has accelerated the development of explainable GNN methods Ying et al. (2019); Pope et al. (2019); Yuan et al. (2021). As illustrated in Fig. 1, explainable GNNs generally identify important subgraphs as the explanation to reflect the logic of the prediction on the input samples.

Although explainable GNN mechanisms enhance the model transparency, the provided explanations can inadvertently leak internal decision logic, thereby increasing the risk of model stealing. In practice, to protect intellectual property, GNN models in critical applications such as drug screening are typically deployed exclusively via online services. As shown in Fig. 1, malicious attackers could query these deployed explainable GNNs to obtain explanations that potentially expose internal decision logic. These explanations may enable attackers to construct a surrogate model that replicates the decision logic of private models. Hence, this raises a concern about the vulnerability of the explainable GNNs to model stealing attacks.

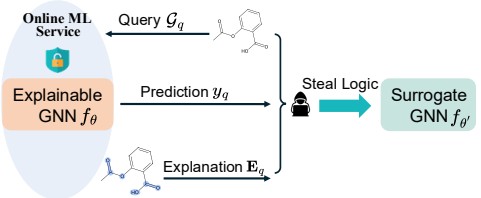

Figure 1: Explainable GNNs and the potential leakage of model logic from explanations.

Previous studies have investigated model stealing attacks against vanilla GNNs, which do not provide explanations for their predictions. Wu *et al.* Wu et al. (2022a) conduct the first investigation on GNN model stealing attacks. Specifically, they train the surrogate model to replicate the predictions of the target model on queried samples. More recent studies further explore strategies such as embedding alignment Shen et al. (2022) and graph augmentation Podhajski et al. (2024); Guan et al. (2024) to

improve attack efficacy. However, these methods generally treat the target model as a black box, ignoring the additional information leaked through explanation mechanisms. For the investigation of model leakage from explanations, early research in computer vision Milli et al. (2019); Yan et al. (2023b) demonstrated that explanations can reveal information about model parameters. However, the unique graph structures and message-passing mechanisms in GNNs significantly differentiate their decision-making and explanation processes. Thus, previous methods for images are not directly applicable to explainable GNNs. Consequently, it is crucial to specifically investigate model stealing attacks against explainable GNNs.

Two major questions remain to answer for the investigation of model stealing attacks against explainable GNNs. *First*, can explanations provided by explainable GNNs facilitate the extraction of the target model's prediction logic? While explanations could provide insights into the GNN's reasoning process, converting this information into effective training signals for the surrogate GNN is non-trivial. *Second*, can attackers efficiently perform model stealing against explainable GNNs under limited query budgets? Real-world scenarios often impose strict constraints on the number of allowed queries, which requires methods to maximize the utility of each interaction with the target explainable GNN. Therefore, how to conduct effective model stealing attack against explainable GNNs within limited queries becomes another critical challenge.

To address these challenges comprehensively, we first propose a causal view of model stealing attacks with explanations. This causal analysis highlights the necessity of explanation alignment for effective logic stealing. In addition, the style invariance property identified in Sec. 4.1 indicates that predictions under various style interventions can be directly inferred without additional queries, thus enabling efficient augmentation of the surrogate model's training set. Motivated by this causal perspective, we propose an Explanation-guided GNN stealing framework (EGSteal). Specifically, EGSteal employs a novel rank-based explanation alignment loss to ensure that the surrogate model not only replicates the predictive behavior of the target model but also captures its underlying decision logic. Furthermore, inspired by our causal analysis, EGSteal incorporates explanation-guided data augmentation, which can facilitate effective model stealing under limited query budgets. Our EGSteal verifies the vulnerability of explainable GNNs under model stealing attacks, which could inspire effective defense methods to enhance model security. In summary, our main contributions are:

- We focus on a novel problem of model stealing attacks on explainable GNNs. In addition, the model stealing with explanations is analyzed from a causal view, which motivates effective solutions.
- We propose a new framework integrating explanation alignment and explanation-guided augmentation to effectively replicate the logic of explainable GNNs under limited query budgets.
- Extensive experiments on various GNN architectures and explanation methods demonstrate that our approach can effectively performs model stealing that captures both prediction behaviors and underlying decision logic.

## 2 RELATED WORKS

**Explainable Graph Neural Networks**. GNNs have demonstrated significant success in many critical domains. To enhance the trustworthiness of GNN predictions, researchers have proposed various explanation methods in the form of important subgraphs. Early methods widely use gradient-based techniques Baldassarre & Azizpour (2019) to identify important features, while CAM Zhou et al. (2016); Pope et al. (2019) generates explanations by examining the final layer's feature maps. Grad-CAM Selvaraju et al. (2017); Pope et al. (2019) further incorporates gradient and CAM to better localize important regions. One notable advancement, GNNExplainer Ying et al. (2019) formulates explanations as a compact subgraph maximizing the mutual information with the model's prediction. PGExplainer Luo et al. (2020) further accelerates this process by training a parameterized explanation generator. Other approaches, such as GraphLIME Huang et al. (2022) adapt local interpretable model-agnostic explanations, and SubgraphX Yuan et al. (2021) employ Monte Carlo tree search. For more related works on explainable GNNs, please refer to Appendix A.1.

**Model Stealing Attacks against GNNs**. Model stealing or model extraction attacks aim to steal the target model information by training a similar surrogate model that matches its prediction behavior Dai et al. (2024); Tramèr et al. (2016); Wu et al. (2022a). Wu et al. Wu et al. (2022a) conduct the first investigation on GNN model stealing attacks by training a surrogate model through queried APIs. Shen et al. Shen et al. (2022) further extend the framework to inductive GNNs, which better reflect real-world scenarios. Building upon this direction, EfficientGNN Podhajski et al. (2024) augments the

shadow dataset with contrastive learning and spectral graph augmentations, and Zhuang et al. Zhuang et al. (2024) recently propose STEALGNN, introducing a more challenging scenario where attackers have no access to any real graph data. With the wide deployment of explainable models, some recent research in computer vision has investigated the risk of explainable models to model logic extraction. For instance, Milli et al. Milli et al. (2019) demonstrate that gradient-based explanations of a model can reveal the model itself. Yan et al. Yan et al. (2023b) further introduce an extra CNN autoencoder to utilize the representations learned by reconstructing the explanations as data augmentations. In this study, we firstly address the vulnerability of explainable GNNs to model stealing attacks. Please refer to Appendix A.2 for more related works on GNN stealing attacks.

## 3 PRELIMINARIES

Let $\mathcal{G} = (\mathcal{V}, \mathcal{E}, \mathbf{X})$ denote a graph, where $\mathcal{V}$ is the set of nodes, $\mathcal{E} \subseteq \mathcal{V} \times \mathcal{V}$ represents the edge set, and $\mathbf{X} \in \mathbb{R}^{|\mathcal{V}| \times d}$ is the node feature matrix with dimension $d$. Let $f_\theta$ denote the target model with parameters $\theta$ trained on $\mathcal{D}_T$. For a explainable target model, the explanation mechanism is represented by $\phi$. In a model stealing attack, the attacker has a shadow dataset containing a handful of query graphs $\mathcal{D}_q$. Given the above notations, we introduce the preliminaries in the following.

**Preliminaries of GNN Explanations**. To integrate both node features and graph topology for representation learning, Graph Neural Networks (GNNs) generally adopt the message-passing mechanism to update node representations by aggregating the information from their neighborhood nodes. Specifically, the embedding of node $v$ at layer $k$ is updated as:

$$\mathbf{h}_v^{(k)} = \text{AGGREGATE}(\mathbf{h}_v^{(k-1)}, \{\text{MSG}(\mathbf{h}_v^{(k-1)}, \mathbf{h}_u^{(k-1)})\}), \tag{1}$$

where $u \in \mathcal{N}(v)$ denoting its neighborhoods. Various explanation methods have been proposed to enhance the explainability of GNNs. The majority of these methods providing explanations in the form of important subgraphs such as Graph-CAM Pope et al. (2019) and GNNExplainer Ying et al. (2019). Therefore, we focus on subgraph explanations. Formally, given an input graph $\mathcal{G} = (\mathcal{V}, \mathcal{E})$ and a GNN $f_\theta$, an explainer $\phi$ identifies a subgraph $\mathcal{G}_E$ as the explanation. In practice, explanations often take a soft form and can be uniformly represented as a node importance vector:

$$\mathbf{E} = [E_1, \ldots, E_{|\mathcal{V}|}]^\top = \phi(f_\theta, \mathcal{G}) \in \mathbb{R}^{|\mathcal{V}|}, \tag{2}$$

where a higher value of $E_i$ indicates a greater contribution of node $v_i$ to the model's graph classification decision. For more details on GNN explanation methods, please refer to Appendix B.

**Threat Model**. The goal of the adversary is to train a surrogate model that matches both the accuracy and decision boundary of the target model under a limited query budget. For the attacker's knowledge, our attack operates under the black-box setting where the target model's architecture, parameters, and internal explanation mechanism are unknown. The attacker possesses a shadow dataset $\mathcal{D}_S$ and a query budget $Q$ for interactions with the target model. For each query graph $\mathcal{G}_q \in \mathcal{D}_Q$, the attacker receives both the model prediction $\hat{y}_q$ and the explanation output $\mathbf{E}_q$, which is a node importance score vector. For the shadow data, we consider two settings (i) In-distribution setting: where the shadow data $\mathcal{D}_S$ follows the same distribution as the target model's training data $\mathcal{D}_T$; (ii) Cross-distribution setting: $\mathcal{D}_S$ and $\mathcal{D}_T$ are not from the sample distribution but are in the same domain.

## 4 METHODOLOGY

In this section, we first analyze the model stealing with explanations from a causal perspective. The key insight is that explanations partition an input graph into two distinct parts: a causal explanation subgraph $\mathcal{G}_E$ containing essential structures determining model predictions, and a non-causal style subgraph $\mathcal{G}_S$ whose modifications do not affect predictions. The identified causal mechanisms could guide surrogate model training for effective logic extraction. In addition, the causal analysis enables the generation of diverse training samples through interventions on non-causal components.

### 4.1 CAUSAL ANALYSIS OF MODEL STEALING WITH EXPLANATIONS

We adopt a causal perspective to analyze the model stealing with explanations. Specifically, we propose a Structural Causal Model (SCM) Pearl (2009) as illustrated in Figure 2. The SCM includes four key variables: the explanation subgraph $\mathcal{G}_E$, the style subgraph $\mathcal{G}_S$, the input graph $\mathcal{G}$, and the prediction outcome $Y$. Each directed link represents a causal relationship between the variables.

Specifically, we detail the causal relationships as follows: **(i)** $\mathcal{G}_E \rightarrow \mathcal{G} \leftarrow \mathcal{G}_S$. The graph $\mathcal{G}$ is formed of the explanation graphs $\mathcal{G}_E$ and style graph $\mathcal{G}_S$. **(ii)** $\mathcal{G}_E \rightarrow Y$. The explanation subgraph $\mathcal{G}_E$ containing critical decision-related structures, which determines the prediction $Y$, while the style subgraph $\mathcal{G}_S$ has no direct causal effect to the prediction.

Following the principle of Independence of Mechanisms, inter-ventions on the style variable $\mathcal{G}_S$ should not change the condi-tional distribution $p(Y|\mathcal{G}_E)$. Formally, for the target model, the following invariance property holds:

$$p^{do(\mathcal{G}_S=\mathcal{G}_S^i)}(Y|\mathcal{G}_E;\theta) = p(Y|\mathcal{G}_E;\theta), \quad \forall \mathcal{G}_S^i \in \mathbb{G}_S \quad (3)$$

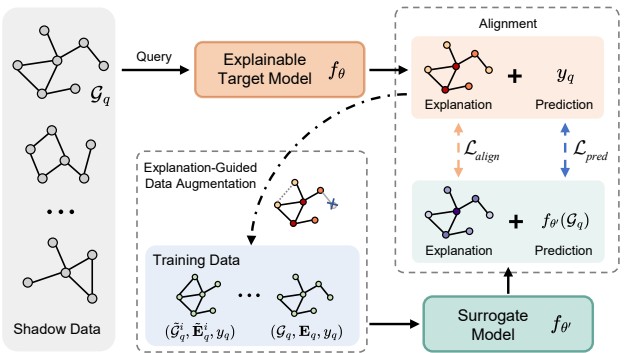

Figure 2: Structural causal model for explainable graph prediction.

where $p^{do(\mathcal{G}_S=\mathcal{G}_S^i)}$ denotes the distribution obtained by intervening to set the style subgraph variable $\mathcal{G}_S$ to a particular value $\mathcal{G}_S^i$, and $\mathbb{G}_S$ represents the set of all possible style subgraph variations. Recall that the objective of model stealing is to learn a surrogate model $f_{\theta'}$ that simulate the behaviors of the target model. Given Eq.(3), the model stealing with explanations should further meet:

$$p^{do(\mathcal{G}_S=\mathcal{G}_S^i)}(Y|\mathcal{G}_E;\theta') = p^{do(\mathcal{G}_S=\mathcal{G}_S^i)}(Y|\mathcal{G}_E;\theta) = p(Y|\mathcal{G}_E;\theta), \quad \forall \mathcal{G}_S^i \in \mathbb{G}_S \quad (4)$$

To satisfy the principle in Eq.(4), the surrogate model $f_{\theta'}$ should fulfill the following two properties:

- **Explanation Alignment**: The explanation graphs generated by the surrogate model $f_{\theta'}$ should align with those of the target model $f_\theta$. This requires explicit alignment of two models.
- **Style Invariance**: Predictions of the surrogate model should remain invariant under style interven-tions. Since $p(Y|\mathcal{G}_E;\theta)$ can be obtained through a single query to the target model, predictions under various style interventions can be determined without additional queries. Consequently, this property enables augmentation of the surrogate model's training set without extra query costs.

### 4.2 FRAMEWORK OF EGSTEAL

In this subsection, we elaborate on EGSteal, which is specifically de-signed to satisfy the critical proper-ties derived from our causal analysis of model stealing with explanations. As illustrated in Fig. 3, our proposed framework comprises two major com-ponents: the logic stealing via expla-nation alignment and the explanation-guided data augmentation. In partic-ular, motivated by the principle of ex-planation alignment, we introduce a novel rank-based explanation align-ment loss, explicitly ensuring that the

Figure 3: Overview framework of our EGSteal.

surrogate model not only replicates the predictive behavior of the target model but also faithfully captures its underlying reasoning patterns. Moreover, guided by the principle of style invariance, EGSteal deploys an explanation-guided data augmentation to perform style interventions. This allows diverse training data generation without additional queries, which enables effective model stealing with limited query budgets.

#### 4.2.1 LOGIC STEALING VIA EXPLANATION ALIGNMENT

Explanations explicitly reveal a model's underlying decision logic. While replicating predictions is a fundamental objective of model stealing, capturing the reasoning process is equally crucial. The necessity of aligning explanations is further justified by the causal analysis presented in Sec. 4.1. Therefore, we propose an explanation alignment module detailed below.

**Explanation Mechanism for Surrogate Model**. To enable explanation alignment, both target and surrogate models must have interpretable explanation mechanisms. While various explanation methods, such as GNNExplainer Ying et al. (2019), could potentially serve this purpose, we specif-ically adopt Graph-CAM Baldassarre & Azizpour (2019); Pope et al. (2019) for two key reasons:

(i) Graph-CAM efficiently obtains node importance scores without requiring additional training of explainer models, thereby efficiently and faithfully reflecting the surrogate model's decision logic; (ii) explanations generated by Graph-CAM naturally support gradient-based optimization, allowing direct backpropagation of explanation alignment losses into the surrogate model's parameters.

Graph-CAM aims to interpret GNN predictions by identifying nodes that contribute significantly to classifications. It leverages node-level embeddings produced by the GNN encoder and traces the prediction process back to these embeddings using classifier weights. Formally, given an input graph with node features $\mathbf{X}$ and adjacency matrix $\mathbf{A}$, the GNN encoder computes node-level embeddings $F_{k,v}(\mathbf{X}, \mathbf{A})$, where $k$ indexes feature dimensions and $v$ indexes nodes. Then the node importance score of node $v$ in the surrogate model $f_{\theta'}$ are given by:

$$E'_v = \sum_k w_k^c \cdot F_{k,v}(\mathbf{X}, \mathbf{A}), \tag{5}$$

where $w_k^c$ represents the classifier weight for feature dimension $k$ and the predicted class $c$. This approach backpropagates class-specific weights to the pre-pooling node features, revealing each node's contribution to the final prediction. Nodes with higher importance scores thus have a stronger influence on the model's decision for the corresponding class.

**Rank-Based Explanation Alignment Loss**. Given the explanations from both surrogate and target models, a natural approach to explanation alignment is to directly minimize the difference between their node importance scores, for instance by applying Mean Squared Error (MSE) loss. However, such a straightforward approach overlooks a key challenge: raw importance scores produced by different models or explanation methods often vary widely in scale and distribution, limiting the effectiveness of direct alignment. To address this issue, we propose aligning explanations based on relative node rankings rather than absolute importance scores. The rationale is that, despite variations in magnitude, the relative ordering of node importance tends to remain stable and effectively reflects the underlying reasoning process.

Following this intuition, we propose a rank-based explanation alignment loss inspired by RankNet Burges et al. (2005). For any pair of nodes $(i, j)$ in a graph, the importance ranking given by the target model is denoted as $r_{ij}$, where $r_{ij} = 1$ if node $i$ is ranked higher (more important) than node $j$ according to the target model's explanations, and $r_{ij} = 0$ otherwise. We compute the difference of their importance scores in the surrogate model as $\Delta_{ij} = E'_i - E'_j$. Based on this difference, the rank-based explanation alignment loss can be computed by:

$$\mathcal{R}(\Delta_{ij}, r_{ij}) = -r_{ij} \log \sigma(\Delta_{ij}) - (1 - r_{ij}) \log(1 - \sigma(\Delta_{ij})), \tag{6}$$

where $\sigma$ is the sigmoid function. To account for all pairwise relationships in the graph, we define the final alignment loss as the average ranking loss over all node pairs:

$$l_{\text{align}} = \frac{1}{|\mathcal{P}|} \sum_{(i,j) \in \mathcal{P}} \mathcal{R}(\Delta_{ij}, r_{ij}), \tag{7}$$

where $\mathcal{P}$ denotes the set of all node pairs with $i < j$ to avoid duplicate comparisons. By explicitly enforcing node-ranking consistency, this alignment ensures that the surrogate model robustly captures the internal decision logic of the target model.

### 4.2.2 EXPLANATION-GUIDED DATA AUGMENTATION

Recall from the causal analysis of model stealing with explanations (Eq. 4) that the surrogate model should satisfy the following style invariance property: $p^{do(\mathcal{G}_S = \mathcal{G}'_S)}(Y|\mathcal{G}_E; \theta') = p(Y|\mathcal{G}_E; \theta)$, where $\mathcal{G}'_S \in \mathbb{G}_S$ represents any modified style subgraph. This property implies that predictions should remain invariant for samples generated through interventions on style subgraphs. We can directly infer labels of intervened samples without additional queries to the target model. Since we do not have access $\mathbb{G}_S$, to simulate style variability, we conduct explanation-guided data augmentations as interventions on the style graph. Specifically, given a graph $\mathcal{G}$ and node importance scores $\mathbf{E}$ obtained from the target explainable GNN $f_\theta$, we first identify nodes with low importance as belonging to the non-causal style subgraph:

$$\mathcal{V}_S = \{v_i \in V \mid \text{rank}(\mathbf{E}_i) \leq \alpha \cdot |\mathcal{V}|\}, \tag{8}$$

where $\mathbf{E}_i$ denotes the importance score of node $i$, and $\alpha \in [0, 1]$ is the proportion of nodes considered least important. We then perform the following explanation-guided data augmentation strategies:

- **Style Graph Node Dropping**: It randomly removes nodes in $\mathcal{V}_S$ to produce an augmented graph: $\tilde{\mathcal{G}} = \mathcal{G} \backslash \text{RandomSelect}(\mathcal{V}_S, \beta \cdot |\mathcal{V}_S|)$, where $\text{RandomSelect}(\mathcal{V}_S, \beta \cdot |\mathcal{V}_S|)$ denotes randomly selecting a fraction $\beta \in [0, 1]$ of nodes from the identified style node set $\mathcal{V}_S$.
- **Style Graph Edge Perturbation**: This strategy randomly modifies edges connecting the nodes in $\mathcal{V}_S$. It will create intervened graph $\tilde{\mathcal{G}}$ with diverse style variations while preserving the causal structure represented by the high-importance nodes.

Using the above explanation-guided data augmentation strategies, each queried graph $\mathcal{G}_q \in \mathcal{D}_Q$ can be augmented into multiple intervened graphs. As for each queried tuple $(\mathcal{G}_q, \mathbf{E}_q, y_q)$ from the target model, it can be augmented into $\{(\tilde{\mathcal{G}}_q^i, \tilde{\mathbf{E}}_q^i, \tilde{y}_q)\}_{i=1}^K$. Specifically, the prediction and explanation of each augmented sample $(\tilde{\mathcal{G}}_q^i, \tilde{\mathbf{E}}_q^i, y_q)$ are derived as follows:

$$\tilde{y}_q^i = y_q, \quad \tilde{\mathbf{E}}_q^i = \mathbf{E}_q[\tilde{\mathcal{G}}_q^i], \tag{9}$$

$\mathbf{E}_q[\tilde{\mathcal{G}}_q^i]$ represents the subset of the original explanation vector obtained by directly omitting nodes removed from the original graph $\mathcal{G}_q$ to construct $\tilde{\mathcal{G}}_q^i$. As suggested in Eq.(9), the prediction $\tilde{y}_q$ on $\tilde{\mathcal{G}}_q^i$ remains $y_q$ for each augmented graph, which in line with the style invariance property. For the corresponding explanation $\tilde{\mathbf{E}}_q^i$, we directly omit nodes removed in $\tilde{\mathcal{G}}_q^i$. This is justified by the assumption that modifications of non-causal style graphs do not alter the relative importance rankings of the remaining causal nodes.

### 4.2.3 FINAL OBJECTIVE FUNCTION

Our final objective function combines explanation alignment and explanation-guided data augmentation. Let $\mathcal{D}_Q$ and $\mathcal{D}_A$ denotes the queried dataset augmented dataset, respectively, the final objective function of EGSteal of optimizing the surrogate model's parameters $\theta'$ can be written as:

$$\min_{\theta'} \mathcal{L}_{\text{total}} = \sum_{\mathcal{G}_i \in \mathcal{D}_Q \cup \mathcal{D}_A} l_{\text{pred}}(y_i, f_{\theta'}(\mathcal{G}_i)) + \lambda l_{\text{align}}(\mathbf{E}_i, \mathbf{E}_i'), \tag{10}$$

where $l_{\text{pred}}$ is the cross-entropy loss, $l_{\text{align}}$ is the rank-based explanation alignment loss in Eq. 7 that aligns the node importance vectors $\mathbf{E}_i$ from the target model and $\mathbf{E}_i'$ from the surrogate model, and $\lambda$ balances the contributions of two loss terms. The detailed training algorithm and time complexity analysis can be found in Appendix C and Appendix D.

## 5 EXPERIMENTS

In this section, we conduct experiments to answer the following research questions:

- **RQ1** How effective is our framework at extracting both prediction behavior and underlying decision logic from explainable GNNs compared to existing model stealing methods?
- **RQ2** How does our method perform under different practical scenarios including distribution shifts between shadow and target datasets, and low explanation qualities?
- **RQ3** How well does our framework generalize across different node classification tasks, model architectures, and GNN explainers?

### 5.1 EXPERIMENTAL SETUP

**Datasets, Metrics, and Implementation Details.** We conduct experiments on seven molecular graph datasets Morris et al. (2020) Hu et al. (2019), including NCI1, NCI109, Mutagenicity, AIDS, ogbg-molhiv, Tox21, and BACE. Each dataset is split into three parts: 40% for training the target model, 20% as the test set, and 40% as the shadow dataset, and the query data with different budgets is sampled from the shadow dataset. The target model uses Graph-CAM as the default explainer. We evaluate our framework using three metrics: (1) Area Under the ROC Curve (AUC) on the test set to assess predictive performance, (2) Prediction Fidelity, which measures the agreement between surrogate and target model predictions, and (3) Rank Correlation of explanations, which quantifies the alignment of decision logic through Kendall's tau coefficient between explanations from both models. Metric formulations and implementation details are provided in Appendix E.1 and E.2.

Table 1: Performance comparison with target models trained from scratch

| Dataset | Metric (%) | Target | TS | MEA-GNN | GNNStealing | EfficientGNN | MRME | DET | STEALGNN | Ours |
|---|---|---|---|---|---|---|---|---|---|---|
| NCI1 | AUC | 81.70 | 74.13 ± 2.57 | 72.29 ± 1.00 | 77.42 ± 2.01 | 75.93 ± 1.32 | 74.25 ± 2.50 | 71.49 ± 1.12 | 69.07 ± 2.75 | **80.74 ± 1.18** |
| | Fidelity | – | 76.23 ± 1.86 | 78.71 ± 1.45 | 80.41 ± 1.16 | 77.42 ± 2.94 | 76.55 ± 1.06 | 71.31 ± 1.43 | 66.27 ± 2.33 | **87.78 ± 0.54** |
| | Rank Corr. | – | 15.22 ± 1.09 | 11.37 ± 1.05 | 18.62 ± 0.53 | 14.90 ± 1.23 | 15.35 ± 0.81 | 9.60 ± 1.30 | 8.82 ± 1.68 | **42.38 ± 1.01** |
| NCI109 | AUC | 80.20 | 71.13 ± 3.54 | 71.11 ± 1.07 | 75.22 ± 0.83 | 73.94 ± 0.87 | 72.16 ± 2.05 | 71.19 ± 0.83 | 70.24 ± 0.90 | **78.08 ± 0.87** |
| | Fidelity | – | 74.40 ± 5.00 | 69.02 ± 5.13 | 76.70 ± 3.90 | 76.51 ± 2.82 | 75.73 ± 3.58 | 70.08 ± 2.15 | 69.26 ± 2.23 | **85.99 ± 1.20** |
| | Rank Corr. | – | 12.91 ± 1.69 | 9.56 ± 0.85 | 14.93 ± 1.65 | 13.91 ± 1.30 | 13.71 ± 1.11 | 8.29 ± 1.26 | 10.27 ± 1.16 | **35.58 ± 0.33** |
| AIDS | AUC | 94.19 | 89.77 ± 1.10 | 88.15 ± 1.29 | 90.66 ± 1.56 | 89.34 ± 1.85 | 89.79 ± 1.10 | 85.49 ± 2.34 | 84.17 ± 1.41 | **93.38 ± 0.54** |
| | Fidelity | – | 87.65 ± 1.69 | 88.25 ± 2.43 | 86.70 ± 1.89 | 87.00 ± 2.85 | 87.60 ± 1.70 | 83.95 ± 1.71 | 79.70 ± 1.98 | **93.25 ± 0.52** |
| | Rank Corr. | – | 35.60 ± 1.86 | 23.97 ± 2.37 | 35.75 ± 2.57 | 36.60 ± 2.07 | 35.67 ± 1.84 | 13.31 ± 6.06 | 30.68 ± 1.82 | **62.91 ± 0.91** |
| Mutagenicity | AUC | 82.57 | 79.21 ± 1.95 | 79.80 ± 0.78 | 81.72 ± 0.41 | 79.38 ± 0.94 | 79.32 ± 2.03 | 79.86 ± 1.99 | 81.17 ± 1.14 | **82.20 ± 1.24** |
| | Fidelity | – | 80.67 ± 3.32 | 80.25 ± 1.15 | 85.05 ± 1.19 | 81.43 ± 1.30 | 80.81 ± 3.27 | 80.14 ± 1.69 | 77.51 ± 1.54 | **88.81 ± 0.88** |
| | Rank Corr. | – | 22.47 ± 3.75 | 21.25 ± 1.81 | 29.33 ± 1.75 | 21.76 ± 1.05 | 22.62 ± 3.52 | 21.85 ± 2.38 | 22.08 ± 1.48 | **44.96 ± 0.41** |

Table 2: Performance comparison with pre-trained target models

| Dataset | Metric (%) | Target | TS | MEA-GNN | GNNStealing | EfficientGNN | MRME | DET | STEALGNN | Ours |
|---|---|---|---|---|---|---|---|---|---|---|
| HIV | AUC | 78.94 | 59.71 ± 3.43 | 61.66 ± 2.80 | 60.25 ± 3.11 | 62.69 ± 1.56 | 61.73 ± 1.73 | 56.30 ± 2.47 | 58.39 ± 3.89 | **66.00 ± 1.76** |
| | Fidelity | – | 95.04 ± 2.35 | 97.34 ± 0.34 | **97.56 ± 0.00** | 97.19 ± 0.40 | 96.02 ± 1.44 | 97.55 ± 0.01 | 97.54 ± 0.02 | 95.54 ± 1.26 |
| | Rank Corr. | – | 2.54 ± 7.54 | 2.89 ± 4.62 | 15.93 ± 4.39 | 2.70 ± 2.61 | 8.14 ± 3.12 | −9.26 ± 8.69 | −8.32 ± 24.92 | **37.42 ± 1.21** |
| Tox21 | AUC | 83.69 | 75.57 ± 1.27 | 70.17 ± 0.60 | 75.06 ± 1.29 | 74.47 ± 1.45 | 75.16 ± 3.85 | 71.46 ± 1.34 | 71.44 ± 4.12 | **76.30 ± 1.88** |
| | Fidelity | – | 86.22 ± 1.88 | 84.95 ± 5.96 | **89.38 ± 0.03** | 81.80 ± 8.34 | 86.87 ± 2.44 | 89.26 ± 0.24 | 89.39 ± 0.02 | 88.53 ± 1.02 |
| | Rank Corr. | – | 4.15 ± 5.74 | −4.04 ± 5.11 | 11.95 ± 3.15 | 2.75 ± 3.15 | 10.52 ± 4.65 | −21.19 ± 2.20 | 19.53 ± 5.31 | **45.01 ± 3.35** |
| BACE | AUC | 87.08 | 66.37 ± 5.73 | 71.68 ± 4.09 | 71.12 ± 6.77 | 69.94 ± 3.90 | 70.34 ± 3.88 | 63.70 ± 4.09 | 57.70 ± 2.07 | **74.15 ± 3.02** |
| | Fidelity | – | 58.67 ± 5.07 | 60.66 ± 8.65 | 60.66 ± 6.13 | 60.79 ± 8.48 | 62.38 ± 3.09 | 56.07 ± 4.93 | 49.57 ± 2.10 | **64.88 ± 8.61** |
| | Rank Corr. | – | 0.53 ± 0.87 | 1.73 ± 1.12 | 3.47 ± 4.80 | −0.74 ± 3.77 | 4.09 ± 4.00 | −1.77 ± 1.95 | −4.00 ± 2.27 | **12.42 ± 2.81** |

**Compared Methods.** We compare our Explanation-Guided GNN Extraction Attack with three categories of methods: (i) the fundamental **teacher-student (TS)** model extraction baseline via prediction APIs Tramèr et al. (2016); (ii) recent works on model stealing attacks targeting GNNs, including **MEA-GNN** Wu et al. (2022a), **GNNStealing** Shen et al. (2022), **EfficientGNN** Podhajski et al. (2024), and **STEALGNN** Zhuang et al. (2024); (iii) methods from other domains that investigate explanation-guided model extraction attacks, specifically **MRME** Milli et al. (2019) and **DET** Yan et al. (2023b). More details on these methods can be found in Appendix E.3.

## 5.2 MODEL EXTRACTION PERFORMANCE

To address **RQ1**, we evaluate our explanation-guided framework against baseline methods across two experimental settings, as shown in Tables 1 and 2.

**Results with Target Models Trained from Scratch.** Table 1 presents results on molecular datasets where GIN target models are trained from scratch. For all experiments in this setting, the budget for querying shadow data points is set to 30% of the target model's training data size. Our method consistently outperforms all baselines across datasets, approaching the target model's performance with smaller gaps. A phenomenon emerges from the metrics: conventional approaches achieve reasonable prediction performance but struggle with explanation alignment. Our framework shows higher explanation correlation scores, with notable improvements over the strongest baselines. This suggests that explanation-guided learning helps the surrogate model better capture the target model's decision logic, rather than merely mimicking its output predictions. Additional results with GCN and GAT target models are presented in Appendix G.1, showing consistent performance improvements across different target model architectures. To further validate the robustness of our approach, we analyze the performance under varying query budgets (10% to 50% of the target model's training data size, detailed in Appendix G.2), which demonstrates that our method consistently outperforms baselines across different query ratios.

**Results with Pre-trained Target Models.** In Table 2, we examine a more challenging scenario using more complex and powerful target models. These models are pre-trained following the strategy in Hu et al. (2019), specifically combining graph-level multi-task supervised learning and node-level self-supervised learning (ContextPred), and then fine-tuned on downstream task datasets. For query data, the budget is set to 30% of the target model's training data size for Tox21 and BACE, but only 10% for HIV due to its larger scale. Tox21 is a multi-label classification dataset, from which we use only the "NR-AhR" label. Additionally, both HIV and Tox21 have class imbalance issues, creating additional extraction challenges. As shown in Table 2, all methods achieve lower overall performance compared to the previous setting, reflecting the increased difficulty. Nevertheless, our framework maintains better performance, particularly in AUC and explanation correlation metrics, suggesting that incorporating the target model's explanation information contributes to improved extraction.

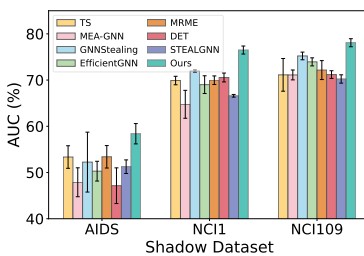 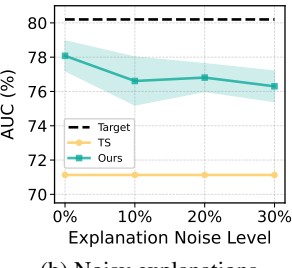 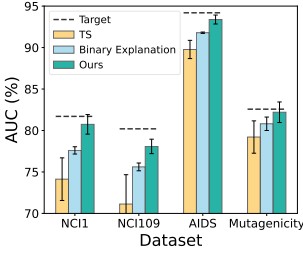

(a) Cross-distribution robustness  (b) Noisy explanations  (c) Binary explanations

Figure 4: Robustness analysis under different practical scenarios. (a) Results when shadow dataset has a distribution shift with training set of target model. (b) Performance under different levels of explanation noises on NCI109 dataset. (c) Results with binary explanations.

## 5.3 ROBUSTNESS ANALYSIS

To address **RQ2**, we evaluate our method's performance under the practical scenarios of distribution shift of shadow dataset, low quality explanations. and explanations in binary mask.

**Cross-Distribution Robustness.** We further examine data distribution shifts by training the target model on NCI109, while using three different shadow datasets: NCI109 (same distribution), NCI1 (MMD=0.016), and AIDS (MMD=0.343). All the models are tested on the NCI109 test set. Figure 4a shows our approach maintains better performance than TS as distribution divergence increases. With NCI1 shadow data, our method achieves higher AUC and explanation rank correlation. This advantage is preserved even with the more challenging AIDS dataset, where our method achieves a 6.68% explanation correlation compared to TS's approximately zero correlation (-0.77%). While extraction performance decreases with increasing distribution shift, our explanation-guided approach demonstrates consistently improved robustness. Additional analysis and implementation details are provided in Appendix G.3 and Appendix F, respectively.

**Robustness to Noisy Explanations.** We simulate unreliable explanation scenarios by randomly shuffling a fraction of the target model's explanation outputs, disrupting the original node importance ranking. Figure 4b shows the AUC performance under different noise levels (0%, 10%, 20%, 30%) on NCI109. Our method demonstrates graceful performance degradation with increasing noise but still outperforms the TS baseline. This suggests that our method exhibits robustness to explanation noise. More results are provided in Appendix G.4.

**Adaptation to Binary Explanations.** Some scenarios only provide coarse-grained explanations, such as binary indicators distinguishing important from irrelevant substructures. Figure 4c shows that our method maintains advantages over TS baselines with this simplified format across all datasets. While performance naturally decreases compared to fine-grained explanations, the framework remains effective across different explanation granularities. Complete results for all metrics and additional analysis are provided in Appendix G.4.

## 5.4 FLEXIBILITY OF EGSTEAL

To answer the **RQ3**, we first conduct experiments of stealing GNN model for node classification. Then, we investigate the impacts of model architectures and explanation mechanisms to EGSteal.

**Flexibility to Node Classification Tasks.** Beyond graph-level tasks, we evaluate our framework on node classification using PubMed and ogbn-arxiv datasets with the same data splitting strategy as in Section 5.1, with the query budget set to 10% of the target model's training data size. To adapt node classification to our framework, we extract 2-hop neighborhood subgraphs centered on each node to transform the task into graph classification. Table 3 shows that our method outperforms baseline approaches across most evaluation metrics, demonstrating effective generalization across different graph learning tasks.

Table 3: Performance comparison on node classification tasks

| Dataset | Metric (%) | Target | TS | MEA-GNN | GNNStealing | EfficientGNN | MRME | DET | STEALGNN | Ours |
|---|---|---|---|---|---|---|---|---|---|---|
| PubMed | AUC | 90.42 | 85.55 ± 0.60 | 86.10 ± 0.63 | 83.86 ± 0.83 | 85.93 ± 0.78 | 85.54 ± 0.61 | 75.38 ± 3.65 | 86.04 ± 0.08 | **87.51 ± 0.93** |
| | Fidelity | – | 82.54 ± 3.02 | 83.56 ± 1.00 | 75.54 ± 0.64 | 83.68 ± 1.02 | 83.06 ± 2.36 | 66.92 ± 3.01 | 86.26 ± 0.73 | **86.66 ± 0.51** |
| | Rank Corr. | – | 33.29 ± 1.40 | 22.66 ± 0.97 | 12.82 ± 2.66 | 34.51 ± 1.47 | 33.30 ± 1.46 | 17.57 ± 4.41 | 36.00 ± 0.59 | **48.94 ± 0.87** |
| ogb-arxiv | AUC | 92.77 | 87.04 ± 0.93 | 88.83 ± 0.49 | 86.35 ± 0.34 | 86.95 ± 0.55 | 87.33 ± 0.75 | 82.58 ± 1.40 | 88.63 ± 0.40 | **89.76 ± 1.21** |
| | Fidelity | – | 61.35 ± 2.45 | 66.62 ± 1.43 | 62.92 ± 1.88 | 63.97 ± 3.26 | 63.01 ± 1.66 | 51.06 ± 2.06 | 66.06 ± 0.85 | **67.30 ± 2.10** |
| | Rank Corr. | – | 26.21 ± 3.54 | 22.23 ± 1.52 | 2.99 ± 2.81 | 30.60 ± 5.14 | 26.44 ± 3.54 | 40.04 ± 1.81 | 36.50 ± 3.25 | **71.27 ± 0.70** |

**Flexibility to Target/Surrogate Model Architectures**. We investigate how different GNN architectures affect extraction performance by examining five architectures (GIN, GCN, GAT, GraphSAGE (GSAGE) and Graph Transformer (GT.) Rampášek et al. (2022) . Figure 5 shows the AUC gains of our method over the TS baseline on NCI109 dataset. Results reveal that GIN as the surrogate model generally achieves better AUC and fidelity gains across different target architectures, likely due to its strong representational capacity. Conversely, when GT. serves as the target, extraction performance gains are relatively lower, suggesting transformer-based models pose greater challenges for extraction. The most significant benefits appear in explanation correlation, where our method shows substantial improvements across almost all architectural combinations. Results for all metrics are provided in Appendix G.5.

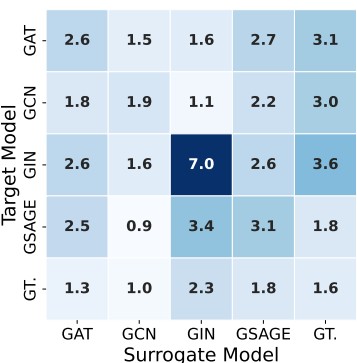

Figure 5: AUC gains (%) of EGSteal over TS using different architectures.

**Flexibility to Explainers**. We assess our framework's performance across five explanation methods applied to the target model: Graph-CAM, Grad, Grad-CAM, GNNExplainer, and PG-Explainer. The experiments are conducted on NCI109 with the same setting as in Section 5.2. For PGEx-

Table 4: Performance of different explanation mechanisms. Gray numbers show performance gains over the TS baseline.

| Explainer | AUC (%) | Fidelity (%) | Rank Corr. (%) |
|---|---|---|---|
| Graph-CAM | $78.08 \pm 0.87$ ↑6.95 | $85.99 \pm 1.20$ ↑11.59 | $35.58 \pm 0.33$ ↑22.67 |
| Grad | $75.38 \pm 1.08$ ↑4.25 | $78.50 \pm 0.94$ ↑4.10 | $52.25 \pm 1.01$ ↑48.40 |
| Grad-CAM | $77.38 \pm 1.10$ ↑6.25 | $85.43 \pm 2.30$ ↑11.03 | $36.60 \pm 0.79$ ↑23.68 |
| GNNExpl. | $75.12 \pm 1.59$ ↑3.99 | $77.75 \pm 1.63$ ↑3.35 | $20.25 \pm 0.62$ ↑22.76 |
| PGExpl. | $75.48 \pm 1.28$ ↑4.35 | $77.99 \pm 2.31$ ↑3.59 | $64.07 \pm 0.79$ ↑65.03 |

plainer, which generates edge importance scores, we convert them to node importance by averaging the scores of all connected edges for each node. Table 4 shows that Graph-CAM and Grad-CAM achieve the highest AUC and fidelity improvements, likely due to alignment with our surrogate model's Graph-CAM based explanation mechanism. PGExplainer demonstrates the strongest explanation correlation gain, suggesting edge-based explanations can be effectively incorporated into our framework. All methods outperform the TS baseline, confirming our approach generalizes well across different explanation techniques. Results for additional datasets are provided in Appendix G.6.

## 5.5 ABLATION STUDY

In this subsection we conduct an ablation study by removing data augmentation and explanation alignment from our framework. Figure 6 shows the AUC performance across four datasets, with complete results available in Appendix G.7. Our experiments demonstrate that both components enhance the framework's performance. Explanation alignment plays a crucial role in transferring decision logic, with its removal causing the most substantial performance drop. Data augmentation primarily improves predictive accuracy, leading to higher AUC and fidelity scores. When both components are removed, the framework essentially reverts to the TS baseline.

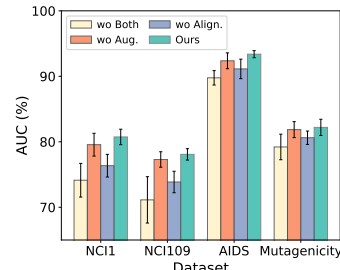

Figure 6: Ablation Study.

## 6 CONCLUSION

This paper investigated the security vulnerabilities of explainable GNNs to model stealing attacks. We proposed EGSteal, a novel stealing framework that exploits explanation information through two complementary components: a rank-based explanation alignment mechanism that effectively captures the target model's decision logic, and an explanation-guided data augmentation strategy that preserves essential causal patterns while enabling efficient training sample generation. Experiments across molecular datasets demonstrated that EGSteal consistently outperforms existing approaches in capturing both predictive behavior and underlying decision logic. These findings reveal an important security-transparency trade-off in the deployment of explainable GNNs in high-stakes domains such as drug discovery. Future work will extend our investigation to more graph learning tasks and develop effective defense mechanisms against explanation-based model stealing attacks.

## ETHICS STATEMENT

This work investigates potential security vulnerabilities in explainable Graph Neural Networks (GNNs) by developing model extraction attacks that exploit explanation information. Our primary objective is to advance the understanding of these vulnerabilities in explainable AI systems to promote the development of more secure and robust GNN deployments. By demonstrating how explanations can inadvertently leak model information, we aim to inform the community about these risks and encourage the development of appropriate defense mechanisms. We believe this work will contribute to improving the security awareness of explainable GNN systems and foster research into protective measures that can mitigate such vulnerabilities while preserving the benefits of model interpretability.

## REPRODUCIBILITY STATEMENT

To ensure reproducibility of our work, we provide comprehensive implementation details throughout the paper and supplementary materials. The methodology of our proposed EGSteal framework is described in detail in Section 4, including the causal analysis, rank-based explanation alignment mechanism, and explanation-guided data augmentation strategies. Complete experimental settings are provided in Section 5.1, including datasets used, model architectures, training configurations, and evaluation metrics. All baseline methods and their adaptations to our experimental setting are detailed in Appendix E.3. The evaluation metrics formulations and hyperparameter specifications are explicitly described in Appendix E.1 and Appendix E.2. We conducted experiments on seven publicly available molecular graph datasets and two node classification datasets, with data splitting strategies and preprocessing steps clearly specified. Our source code is hosted in an anonymous repository, with the link provided at the end of the Abstract.

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

APPENDIX CONTENTS

# A  DETAILS OF RELATED WORKS

## A.1  EXPLAINABLE GRAPH NEURAL NETWORKS

Graph Neural Networks (GNNs) have demonstrated remarkable success across various domains, including financial analysis Harl et al. (2020); WANG et al. (2022); Lv et al. (2019), recommendation systems Chen et al. (2022); Wu et al. (2022b); Hamilton et al. (2017), and biological analysis Li et al. (2021); Sun et al. (2020); Xiong et al. (2021); Kawahara et al. (2017). The success of GNNs largely relies on their message-passing mechanism, where node representations are iteratively updated by aggregating information from their neighbors Xu et al. (2018); Kipf & Welling (2017); Veličković et al. (2018). This enables GNNs to effectively capture both node features and graph structural information, leading to remarkable performance in various tasks including node classification, link prediction, and graph classification.

To enhance the interpretability of these sophisticated models, researchers have proposed various explanation methods Yuan et al. (2022); Dai & Wang (2021; 2025). Early approaches follow methods in computer vision to use gradient and attribution-based techniques, such as sensitivity analysis and guided backpropagation Baldassarre & Azizpour (2019) utilize gradient information to identify important input features, while Class Activation Mapping (CAM) Zhou et al. (2016); Pope et al. (2019) generates explanations by examining the final convolutional layer's feature maps. Grad-CAM Selvaraju et al. (2017); Pope et al. (2019) further improves upon CAM by incorporating gradient information to better localize important regions in the input graph. These methods, though intuitive, often struggle to capture complex structural dependencies in GNNs.

More sophisticated approaches have been developed to address these limitations. GNNExplainer Ying et al. (2019) formulates explanation generation as an optimization problem that identifies a compact subgraph maximizing the mutual information with the model's prediction. PGExplainer Luo et al. (2020) extends this idea by training a parameterized explanation generator that learns to identify important edges across multiple instances. GraphLIME Huang et al. (2022) adapts the concept of local interpretable model-agnostic explanations to graph data, while SubgraphX Yuan et al. (2021) employs monte carlo tree search and Shapley values to generate hierarchical explanations. Graph-Mask Schlichtkrull et al. (2020) takes a different approach by learning to identify dispensable edges during message passing. These methods have demonstrated increasing effectiveness in providing insights into GNN predictions, though their deployment raises important questions about the potential leakage of model decision logic compared to traditional black-box predictions.

## A.2  MODEL STEALING ATTACKS AGAINST GNNS

As an important aspect of privacy attacks, model stealing or model extraction attacks aim to extract the target model information by learning a surrogate model that behaves similarly to the target model Dai et al. (2024); Tramèr et al. (2016). Generally, the attacker will first query the APIs of the target model to obtain predictions on the shadow dataset. It then leverages the shadow dataset and the corresponding predictions to train the surrogate model for model extraction attack Tramèr et al. (2016). Following Tramèr et al. (2016), recent works have developed various model extraction attacks against graph neural networks. Wu et al. Wu et al. (2022a) proposed the first framework of GNN model extraction attacks along multiple dimensions based on the attacker's different background knowledge of the shadow dataset: access to node attributes, knowledge of graph structure, and availability of shadow graphs. This taxonomy framework helps understand different attack scenarios in real-world applications, though their focus remains on transductive settings where attackers can access training processes. Shen et al. Shen et al. (2022) made a significant advance by proposing the first model-stealing attacks against inductive GNNs, which better reflects real-world deployment scenarios. They increased the amount of information extracted from the victim model by aligning not only the predictions but also other responses such as node embeddings or prediction logits (soft label). Building upon this direction, Podhajski et al. Podhajski et al. (2024) further augmented the node embeddings with graph contrastive learning and spectral graph augmentations, and Zhuang et al. Zhuang et al. (2024) recently proposed STEALGNN, introducing a more challenging scenario where attackers have no access to any real graph data. Their work demonstrates the possibility of model extraction through carefully designed synthetic graph generation.

With the popularity of explainable AI, early research in other fields has demonstrated that explanations can reveal information about model parameters Aïvodji et al. (2020); Miura et al. (2024); Yan et al. (2023a). Milli et al. Milli et al. (2019) demonstrated that gradient-based explanations of a model can reveal the model itself. Their theoretical analysis proved that with gradient information, the number of queries needed to reconstruct a model can be significantly reduced compared to prediction-only approaches. Yan et al. Yan et al. (2023b) further introduced an extra CNN autoencoder to utilize the representations learned by reconstructing the explanations. Other studies explore the possibility of data-free model extraction with explainable AI Yan et al. (2023a). However, the unique graph structures and message-passing mechanisms in GNNs significantly differentiate their decision-making and explanation processes. Thus, previous methods for images are not directly applicable to explainable GNNs. With the widespread adoption of explainable GNNs in critical areas such as drug discovery, with explanation methods like GNNExplainer Ying et al. (2019), it is necessary to address the *security risks of model extraction attacks arising from explanations*. To the best of our knowledge, we are the *first* to investigate the susceptibility of GNNs to model extraction attacks caused by model explanations.

## B  DETAILS OF EXPLANATION APPROACHES

Here we provide the detailed formulations of the explanation methods used in our experiments. Let $\mathbf{F}^{(L)} \in \mathbb{R}^{|\mathcal{V}| \times d}$ denote the node representations from the final layer $L$ of the GNN encoder, where $d$ is the hidden dimension. For any node $v \in \mathcal{V}$, we use $\mathbf{F}^{(L)}_{k,v}$ to denote its $k$-th feature.

**Graph-CAM**  The Graph Class Activation Mapping generates node importance scores in three steps. First, it performs global average pooling on the final layer representations:

$$e_k = \frac{1}{|\mathcal{V}|} \sum_{v \in \mathcal{V}} \mathbf{F}^{(L)}_{k,v}, \tag{11}$$

Then, the class score for class $c$ is computed as:

$$y_c = \sum_k w_k^c e_k, \tag{12}$$

where $w_k^c$ represents the weight connecting the $k$-th feature to class $c$ in the classification layer. Finally, the importance score for node $v$ is obtained by:

$$\mathbf{E}_v = \sum_k w_k^c \mathbf{F}^{(L)}_{k,v} \tag{13}$$

**Gradient-based**  This method computes node importance scores by taking the gradients of the target class score $y_c$ with respect to the node features in the final layer:

$$\mathbf{E}_v = \|\mathrm{ReLU}(\frac{\partial y_c}{\partial \mathbf{F}^{(L)}_v})\|, \tag{14}$$

where $\mathbf{F}^{(L)}_v$ denotes the feature vector of node $v$ at layer $L$.

**Grad-CAM**  The Gradient-weighted Class Activation Mapping first computes class-specific weights by averaging the gradients:

$$\alpha_k^c = \frac{1}{|\mathcal{V}|} \sum_{v \in \mathcal{V}} \frac{\partial y_c}{\partial \mathbf{F}^{(L)}_{k,v}}, \tag{15}$$

The node importance scores are then calculated as:

$$\mathbf{E}_v = \mathrm{ReLU}(\sum_k \alpha_k^c \mathbf{F}^{(L)}_{k,v}) \tag{16}$$

**GNNExplainer and PGExplainer**  For these methods, we use their official implementations. Since PGExplainer generates edge importance scores $\mathbf{E}_{(u,v)}$, we convert them to node importance scores by averaging over the adjacent edges:

$$\mathbf{E}_v = \frac{1}{|\mathcal{N}(v)|} \sum_{u \in \mathcal{N}(v)} \mathbf{E}_{(u,v)}, \tag{17}$$

where $\mathcal{N}(v)$ denotes the neighbors of node $v$.

## C  TRAINING ALGORITHM

We present the complete training procedure for our explanation-guided model stealing framework in Algorithm 1. The algorithm consists of two main phases: data collection with explanation-guided augmentation and model training with explanation alignment. The algorithm first collects predictions and explanations from the target model within the query budget, then generates augmented training samples using our explanation-guided strategies. The surrogate model is trained to minimize the unified objective that combines prediction alignment and explanation consistency.

---

**Algorithm 1** Explanation-Guided Model Stealing

---

**Require:** Target model $f_\theta$, shadow dataset $\mathcal{D}_s$, query budget $Q$
**Ensure:** Trained surrogate model $f_{\theta'}$
 1: // Phase 1: Data Collection and Augmentation
 2: Sample $Q$ graphs $\{\mathcal{G}_q\}_{q=1}^Q$ from $\mathcal{D}_s$
 3: $\mathcal{D}_{\text{train}} \leftarrow \emptyset$
 4: **for** each graph $\mathcal{G}_q$ **do**
 5:     Query $f_\theta$ to obtain $(y_q, \mathbf{E}_q)$
 6:     Generate augmented graphs $\{\tilde{\mathcal{G}}_q^i\}_{i=1}^K$
 7:     $\mathcal{D}_{\text{train}} \leftarrow \mathcal{D}_{\text{train}} \cup \{(\mathcal{G}_q, \mathbf{E}_q, y_q)\} \cup \{(\tilde{\mathcal{G}}_q^i, \tilde{\mathbf{E}}_q^i, y_q)\}_{i=1}^K$
 8: **end for**
 9: // Phase 2: Model Training
10: Initialize $f_{\theta'}$
11: Train $f_{\theta'}$ by minimizing $\mathcal{L}_{\text{total}} = \mathcal{L}_{\text{pred}} + \lambda \mathcal{L}_{\text{align}}$ in Eq. 10
12: **return** $f_{\theta'}$

---

## D  COMPUTATIONAL COMPLEXITY

We analyze the computational complexity of our explanation-guided model stealing framework. Let $Q$ denote the query budget, $|\mathcal{V}|$ and $|\mathcal{E}|$ represent the average number of nodes and edges in each graph, $L$ be the number of GNN layers, $d$ be the hidden dimension of neural networks, and $\alpha$ be the data augmentation ratio.

For the data collection phase, we perform $Q$ queries to the target model, obtaining predictions and explanations with a cost of $O(Q \cdot |\mathcal{V}|)$ for processing the explanations. The subsequent data augmentation generates $\alpha \cdot Q$ additional training samples without requiring further queries. Since each augmentation operation (node dropping, edge perturbation, or subgraph extraction) needs to process the graph structure, it requires $O(|\mathcal{V}| + |\mathcal{E}|)$ operations per graph, resulting in a total augmentation complexity of $O(\alpha \cdot Q \cdot (|\mathcal{V}| + |\mathcal{E}|))$.

During the model training phase, we process a total of $(1 + \alpha)Q$ samples over $T$ iterations. For each sample, the GNN forward and backward propagation requires $O(L \cdot d \cdot (|\mathcal{V}| + |\mathcal{E}|))$ operations, where the term $(|\mathcal{V}| + |\mathcal{E}|)$ accounts for message passing through all nodes and edges. The loss computation involves the cross-entropy prediction loss with $O(c)$ complexity and the ranking-based explanation alignment loss with $O(|\mathcal{V}|^2)$ complexity due to pairwise comparisons of node importance. Considering all $(1 + \alpha)Q$ samples and $T$ iterations, the overall training complexity becomes $O(T \cdot (1 + \alpha)Q \cdot (L \cdot d \cdot (|\mathcal{V}| + |\mathcal{E}|) + c + |\mathcal{V}|^2))$.

The dominant computational factor in our approach is the quadratic term from the explanation alignment loss. While this introduces additional cost compared to standard GNN training, it is essential for effectively capturing the target model's reasoning process, and the overhead remains manageable for typical graph datasets.

## E  EXPERIMENTAL SETTING

### E.1  EVALUATION METRICS DETAILS

We employ three metrics to evaluate the effectiveness of our model stealing framework:

- **Area Under the ROC Curve (AUC)** on the test set, which measures the predictive performance of the surrogate model.

- **Prediction Fidelity**, which measures the agreement between the predictions of the surrogate and target models:

$$\text{Fidelity} = \frac{1}{|\mathcal{D}|} \sum_{\mathcal{G} \in \mathcal{D}} \mathbb{1}[y_{\theta'}(\mathcal{G}) = y_{\theta}(\mathcal{G})], \tag{18}$$

  where $y_{\theta'}(\mathcal{G})$ and $y_{\theta}(\mathcal{G})$ denote the predictions from the surrogate and target models respectively, and $\mathcal{D}$ is the test dataset.

- **Rank Correlation**, which quantifies the alignment of decision logic between the surrogate and target models through the Kendall's tau coefficient:

$$\tau = \frac{1}{|\mathcal{D}|} \sum_{\mathcal{G} \in \mathcal{D}} \text{corr}_{\tau}(\mathbf{E}_{\theta'}(\mathcal{G}), \mathbf{E}_{\theta}(\mathcal{G})), \tag{19}$$

  where $\text{corr}_{\tau}$ denotes the Kendall's tau coefficient, and $\mathbf{E}_{\theta'}(\mathcal{G})$ and $\mathbf{E}_{\theta}(\mathcal{G})$ are explanations from the surrogate and target models for graph $\mathcal{G}$, respectively.

### E.2 MODEL HYPERPARAMETERS AND TRAINING DETAILS

Our target models implemented various architectures including GIN, GCN, GAT, GraphSAGE, and Graph Transformer, while surrogate models consistently used GIN architecture. All models were configured with 3 GNN layers and 128 hidden dimensions, trained using Adam optimizer (learning rate 0.001) with batch size 64. Target models were trained for 200 epochs with checkpoints selected based on validation AUC. As for pretrained target models, we used a 5-layer GIN with 300-dimensional embeddings, fine-tuned for 100 epochs. Our data splitting strategy allocated 40% for target model training or fine-tuning (further split into 80% train, 20% validation), 40% for shadow data, and 20% for test data. All experiments were conducted on a single NVIDIA RTX A6000 GPU and repeated five times with different random seeds [41, 42, 43, 44, 45], with final results reported as means with standard deviations.

### E.3 COMPARED METHODS

Firstly, we include the fundamental teacher-student framework of the model extraction attack:

- **Teacher-Student** Tramèr et al. (2016) is the fundamental model extraction baseline to train a surrogate model via the target model's input data and prediction APIs.

We include recent works on model stealing attacks targeting GNNs:

- **MEA-GNN** Wu et al. (2022a) provides a taxonomy of GNN extraction attacks on transductive node classification tasks, categorized into 7 attacks based on the available knowledge about the target dataset and query graphs. We adapt this framework's Attack-3 to our graph classification setting, where query graphs are known but other training data remains unknown.

- **GNNStealing** Shen et al. (2022) introduces two types of attacks where the attack can obtain the query graphs (type I) and when the graph structural information is missing (type II). They also include the predicted logits (soft labels) in complement to the hard labels to enhance the performance. We can access the whole query graph in our setting, so we adapt their type I attack and use soft logits as a response for our graph classification setting.

- **EfficientGNN** Podhajski et al. (2024) followed GNNStealing and further enriched the learning of node embeddings with graph contrastive learning. Additionally, they introduced a spectral graph augmentation method to increase the stealing efficiency. We adapt their data augmentation strategies for comparison.

- **STEALGNN** Zhuang et al. (2024) introduces a graph generator to generate query graphs for data-free GNN extraction attacks. During training, the graph generator and surrogate model are trained adversarially to generate diverse graphs that better capture the victim model's decision-making process. We adapt their approach to our setting by combining both the generated graphs and our query graphs to train the surrogate model.

We also include recent studies on leveraging explanations for model extraction in other domains to offer a broader perspective:

- **MRME** Milli et al. (2019) explored the possibility of quicker model retrieval through gradient-based explanations. We adapt this method to our GNN setting by directly aligning the gradient-based explanations of graph neural networks between the target and surrogate model.
- **DET** Yan et al. (2023b) introduced an autoencoder to reconstruct the original image and an extra autoencoder reconstruct their explanations. By combining the embeddings of two encoders, they use the features learned from explanations to train a surrogate classifier.

# F   IMPLEMENTATION DETAILS OF CROSS-DISTRIBUTION EXPERIMENTS

## F.1   FEATURE DIMENSION ALIGNMENT

In our cross-distribution experiments, we encounter feature dimension misalignment between different molecular graph datasets. Specifically, while NCI109 and AIDS datasets have 38-dimensional node features, NCI1 has 37-dimensional features. To enable cross-dataset model stealing attacks while preserving the original feature information, we pad the NCI1 features with an additional zero dimension:

$$\mathbf{x}_{NCI1}^{new} = [\mathbf{x}_{NCI1}; 0] \in \mathbb{R}^{38} \tag{20}$$

where $\mathbf{x}_{NCI1} \in \mathbb{R}^{37}$ is the original feature vector and $[\cdot; \cdot]$ denotes feature concatenation. This padding approach maintains the intrinsic feature patterns of NCI1 while ensuring compatible input dimensions for the GNN models.

## F.2   MAXIMUM MEAN DISCREPANCY COMPUTATION

To quantify the distribution differences between datasets, we compute the Maximum Mean Discrepancy (MMD) using a 16-dimensional feature vector that captures key structural properties of each graph. The MMD is defined as:

$$\begin{aligned} \text{MMD}^2(\mathcal{P}, \mathcal{Q}) =& \mathbb{E}_{x,x'\sim\mathcal{P}}[k(x, x')] + \mathbb{E}_{y,y'\sim\mathcal{Q}}[k(y, y')] \\ &- 2\mathbb{E}_{x\sim\mathcal{P},y\sim\mathcal{Q}}[k(x, y)] \end{aligned} \tag{21}$$

where $\mathcal{P}$ and $\mathcal{Q}$ are two distributions, and $k(\cdot, \cdot)$ is a kernel function.

For each graph, we extract the following features:

1) **Degree Distribution Statistics** (5 features): - Mean node degree - Standard deviation of node degrees - 25th percentile of node degrees - Median node degree - 75th percentile of node degrees

2) **Clustering Coefficient Statistics** (5 features): - Mean clustering coefficient - Standard deviation of clustering coefficients - 25th percentile of clustering coefficients - Median clustering coefficient - 75th percentile of clustering coefficients

3) **Graph Diameter** (1 feature): - Diameter of the largest connected component

4) **Spectral Features** (5 features): - Top 5 eigenvalues of the adjacency matrix - For graphs with fewer than 5 nodes, we pad with zeros to maintain fixed dimensionality

After extracting these 16-dimensional feature vectors for all graphs in both datasets, we normalize them using z-score normalization:

$$\mathbf{x}_{normalized} = \frac{\mathbf{x} - \mu}{\sigma + \epsilon} \tag{22}$$

where $\mu$ and $\sigma$ are the mean and standard deviation computed across all graphs from both datasets, and $\epsilon = 10^{-8}$ is added for numerical stability.

We then compute the MMD using a Gaussian kernel:

$$k(x, y) = \exp(-\gamma ||x - y||^2) \tag{23}$$

where $\gamma = 0.5$ is the kernel bandwidth parameter. Using this approach, we compute the MMD between NCI109 and NCI1 datasets, as well as between NCI109 and AIDS datasets, to quantify their distribution differences.

# G ADDITIONAL RESULTS

## G.1 ADDITIONAL RESULTS WITH DIFFERENT TARGET MODEL ARCHITECTURES

To demonstrate the effectiveness of our approach across different target model architectures, we conduct additional experiments with GCN and GAT as target models. All surrogate models use the GIN architecture, and we maintain the same experimental setup as described in Section 5.2.

Table 5: Performance comparison with GCN target models trained from scratch

| Dataset | Metric (%) | Target | TS | MEA-GNN | GNNStealing | EfficientGNN | MRME | DET | STEALGNN | Ours |
|---|---|---|---|---|---|---|---|---|---|---|
| NCI1 | AUC | 74.68 | 73.17 ± 1.33 | 71.89 ± 0.83 | 71.87 ± 1.03 | 72.77 ± 1.00 | 73.23 ± 1.27 | 70.88 ± 0.60 | 71.36 ± 0.69 | **73.64 ± 0.49** |
| | Fidelity | – | 85.35 ± 2.17 | 84.43 ± 1.73 | 83.87 ± 1.51 | 84.94 ± 1.50 | 85.89 ± 2.29 | 80.44 ± 0.90 | 85.56 ± 0.72 | **85.89 ± 1.54** |
| | Rank Corr. | – | 10.68 ± 1.34 | 9.92 ± 3.29 | 10.40 ± 2.63 | 9.42 ± 2.18 | 10.77 ± 1.85 | 6.17 ± 3.69 | 9.46 ± 1.65 | **37.00 ± 0.60** |
| NCI109 | AUC | 72.66 | 71.13 ± 1.08 | 70.76 ± 0.68 | 70.72 ± 1.24 | 70.17 ± 0.86 | 71.11 ± 1.08 | 68.91 ± 1.65 | 69.56 ± 1.03 | **72.22 ± 0.34** |
| | Fidelity | – | 86.88 ± 0.99 | **88.41 ± 0.98** | 88.02 ± 3.32 | 86.40 ± 1.88 | 87.05 ± 0.84 | 80.82 ± 2.11 | 87.47 ± 0.37 | 87.90 ± 0.84 |
| | Rank Corr. | – | 13.58 ± 0.97 | 16.75 ± 1.23 | 8.97 ± 2.62 | 15.41 ± 2.06 | 13.73 ± 0.73 | 11.22 ± 1.34 | 13.86 ± 0.91 | **31.61 ± 0.28** |
| AIDS | AUC | 87.96 | 78.86 ± 4.19 | 80.04 ± 2.09 | 80.24 ± 4.60 | 78.44 ± 4.03 | 79.01 ± 4.29 | 85.49 ± 2.34 | 83.02 ± 0.66 | 84.53 ± 2.11 |
| | Fidelity | – | 91.60 ± 3.17 | 92.20 ± 1.62 | 90.40 ± 0.93 | **92.95 ± 0.53** | 92.65 ± 1.31 | 83.95 ± 1.71 | 89.99 ± 15.41 | 92.20 ± 0.97 |
| | Rank Corr. | – | 11.15 ± 4.56 | 19.17 ± 4.14 | 12.57 ± 6.97 | 12.60 ± 3.73 | 11.38 ± 4.12 | 13.31 ± 6.07 | 26.82 ± 3.48 | **53.82 ± 0.93** |
| Mutagenicity | AUC | 84.00 | 80.01 ± 1.59 | 81.17 ± 0.76 | **82.91 ± 0.56** | 79.69 ± 1.28 | 80.04 ± 1.54 | 79.42 ± 0.97 | 81.19 ± 0.25 | 81.58 ± 0.79 |
| | Fidelity | – | 90.47 ± 1.32 | 89.57 ± 2.40 | **93.66 ± 1.22** | 89.99 ± 1.71 | 90.73 ± 1.22 | 84.57 ± 0.69 | 84.25 ± 5.24 | 92.11 ± 0.79 |
| | Rank Corr. | – | 26.18 ± 1.44 | 24.97 ± 2.85 | 17.04 ± 3.85 | 25.70 ± 1.86 | 26.23 ± 1.39 | 21.11 ± 2.93 | 29.19 ± 0.23 | **45.16 ± 0.98** |

Table 6: Performance comparison with GAT target models trained from scratch

| Dataset | Metric (%) | Target | TS | MEA-GNN | GNNStealing | EfficientGNN | MRME | DET | STEALGNN | Ours |
|---|---|---|---|---|---|---|---|---|---|---|
| NCI1 | AUC | 79.00 | 73.62 ± 1.00 | 72.29 ± 1.01 | 74.68 ± 0.58 | 74.13 ± 1.06 | 73.91 ± 0.95 | 66.59 ± 2.26 | 70.55 ± 1.72 | **76.01 ± 0.44** |
| | Fidelity | – | 82.77 ± 1.10 | 78.71 ± 1.45 | 84.48 ± 1.52 | 84.53 ± 0.24 | 82.87 ± 1.31 | 74.38 ± 2.81 | 79.76 ± 1.39 | **88.37 ± 0.84** |
| | Rank Corr. | – | 14.15 ± 1.04 | 11.38 ± 1.05 | 10.65 ± 2.52 | 15.26 ± 0.90 | 14.34 ± 0.99 | 10.27 ± 4.16 | 12.49 ± 2.09 | **37.76 ± 0.52** |
| NCI109 | AUC | 73.99 | 72.93 ± 1.26 | 73.24 ± 0.68 | 70.23 ± 4.76 | 73.03 ± 0.49 | 73.21 ± 1.23 | 70.08 ± 1.03 | 71.83 ± 0.80 | **74.56 ± 1.00** |
| | Fidelity | – | 80.17 ± 2.76 | 80.00 ± 4.32 | 78.91 ± 9.86 | 83.27 ± 1.65 | 82.16 ± 1.61 | 75.76 ± 1.24 | 80.12 ± 1.40 | **85.99 ± 1.12** |
| | Rank Corr. | – | 10.22 ± 3.72 | 13.35 ± 2.44 | 4.32 ± 3.90 | 11.97 ± 3.13 | 10.92 ± 2.85 | 10.02 ± 3.20 | 12.37 ± 0.67 | **30.50 ± 0.40** |
| AIDS | AUC | 89.89 | 88.05 ± 0.98 | 85.51 ± 1.73 | 86.69 ± 2.49 | 88.23 ± 1.81 | 88.15 ± 1.14 | 85.49 ± 2.34 | 85.12 ± 0.17 | **88.77 ± 0.61** |
| | Fidelity | – | 86.40 ± 1.33 | 85.40 ± 2.59 | 85.50 ± 2.49 | 86.90 ± 1.11 | 86.20 ± 1.56 | 83.95 ± 1.71 | 82.91 ± 0.51 | **88.05 ± 0.70** |
| | Rank Corr. | – | 13.41 ± 3.72 | 20.11 ± 2.66 | 15.54 ± 1.55 | 13.77 ± 1.85 | 13.40 ± 3.69 | 13.31 ± 6.07 | 14.52 ± 3.30 | **46.52 ± 1.35** |
| Mutagenicity | AUC | 84.06 | 80.81 ± 0.97 | 79.85 ± 1.15 | 82.53 ± 0.79 | 81.31 ± 1.38 | 80.83 ± 1.03 | 77.25 ± 1.01 | 81.05 ± 0.25 | **83.16 ± 0.71** |
| | Fidelity | – | 85.40 ± 1.05 | 84.34 ± 1.65 | **89.57 ± 1.43** | 85.54 ± 1.35 | 85.35 ± 1.16 | 76.61 ± 0.99 | 84.74 ± 0.68 | 89.04 ± 1.49 |
| | Rank Corr. | – | 22.54 ± 1.77 | 24.56 ± 1.72 | 19.17 ± 7.48 | 22.65 ± 1.33 | 22.32 ± 1.57 | 18.20 ± 3.24 | 24.60 ± 1.46 | **50.18 ± 0.49** |

## G.2 RESULTS OF VARYING QUERY BUDGETS

To investigate how varying query budgets affect extraction performance, we examine results when varying query budgets from 10% to 50% of the target model's training data size across four molecular datasets. As shown in Figure 7, our approach consistently outperforms baselines across all query budgets, with particularly notable advantages at lower query ratios. This demonstrates that explanation information provides valuable guidance when query access is limited. The performance gap narrows at higher query ratios, but remains meaningful even at 50%. This confirms that our approach makes more efficient use of query opportunities. The consistency of improvements across different datasets suggests that our explanation-guided framework provides practical value, especially in scenarios where query budgets are restricted due to cost, time, or detection avoidance considerations.

## G.3 ADDITIONAL RESULTS ON CROSS-DISTRIBUTION ANALYSIS

Figure 8 presents the fidelity and explanation rank correlation results for cross-distribution extraction performance, complementing the AUC results shown in the main text. The results consistently demonstrate that our explanation-guided approach maintains better robustness across all metrics

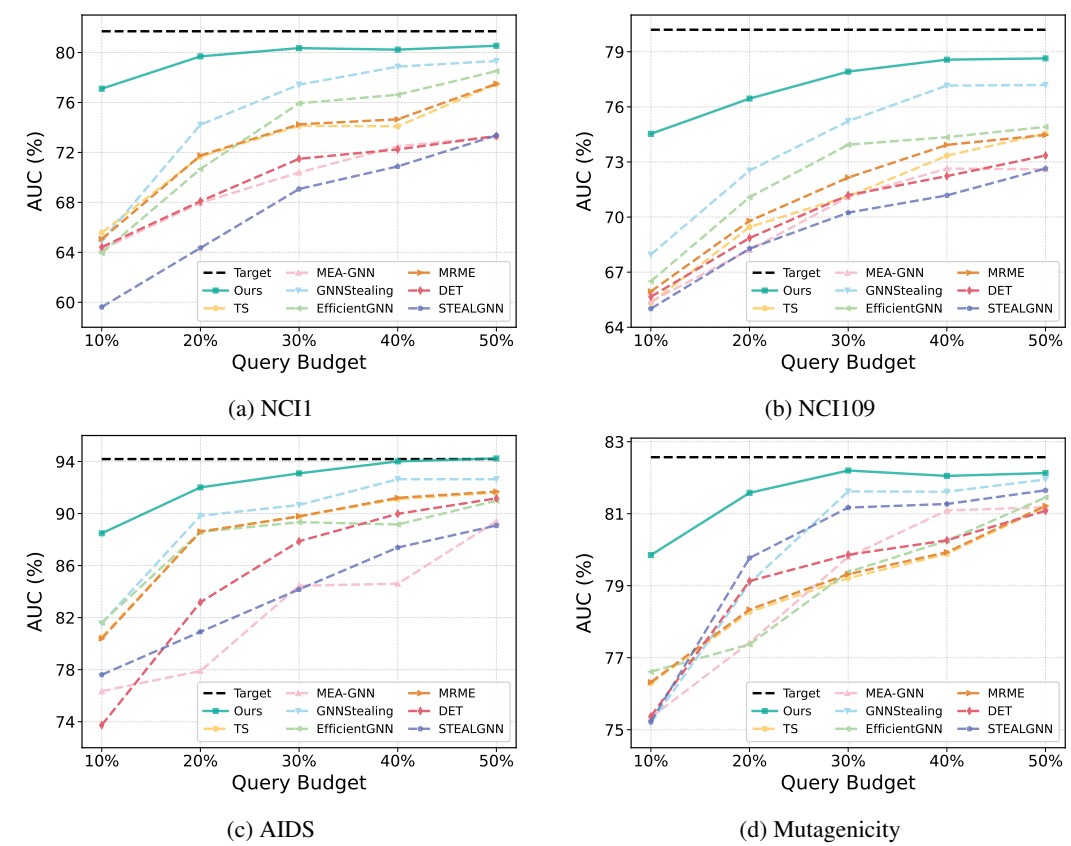

Figure 7: Impact of query budget on extraction performance across datasets. The x-axis represents the percentage of target model's training data size used for querying, and y-axis shows the AUC (%).

as distribution shift increases. The performance gap is particularly notable for explanation rank correlation, which drops more severely for TS as distribution shift increases. This suggests that conventional extraction methods primarily capture surface-level prediction patterns that don't transfer well across distributions, while our approach better captures the underlying decision logic.

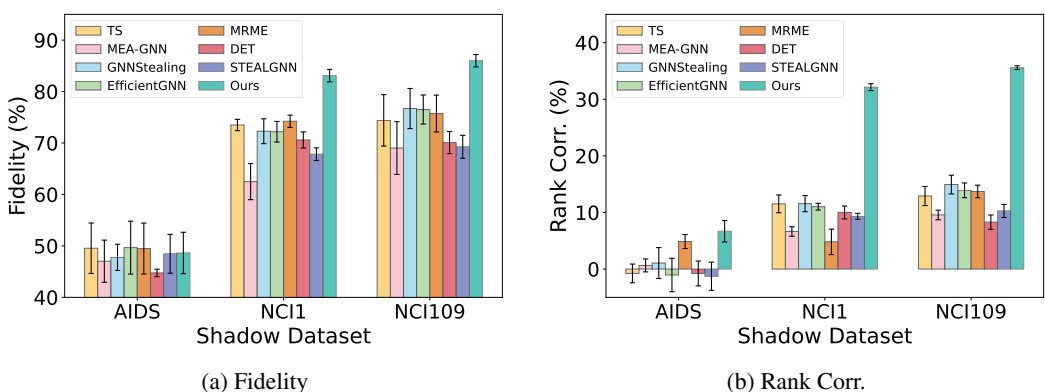

Figure 8: Evaluation of model extraction performance under different shadow dataset distributions. Target model is trained on NCI109.

## G.4 ADDITIONAL RESULTS ON EXPLANATION QUALITY ANALYSIS

**Complete Results for Noisy Explanations** Figure 9 presents the complete performance evaluation under different explanation noise levels across all three metrics. The results demonstrate consistent degradation patterns as noise increases from 0% to 30%, with all metrics showing gradual performance decline and our method's performance approaching that of the TS baseline at higher noise levels. The explanation rank correlation metric exhibits the most significant sensitivity to noise, which is expected since our rank-based alignment mechanism relies on the relative ordering of node importance. However, even with 30% noise injection, our method maintains meaningful improvements over the TS baseline across all datasets. The fidelity metric shows more resilience to noise compared to explanation correlation, while AUC performance demonstrates intermediate sensitivity.

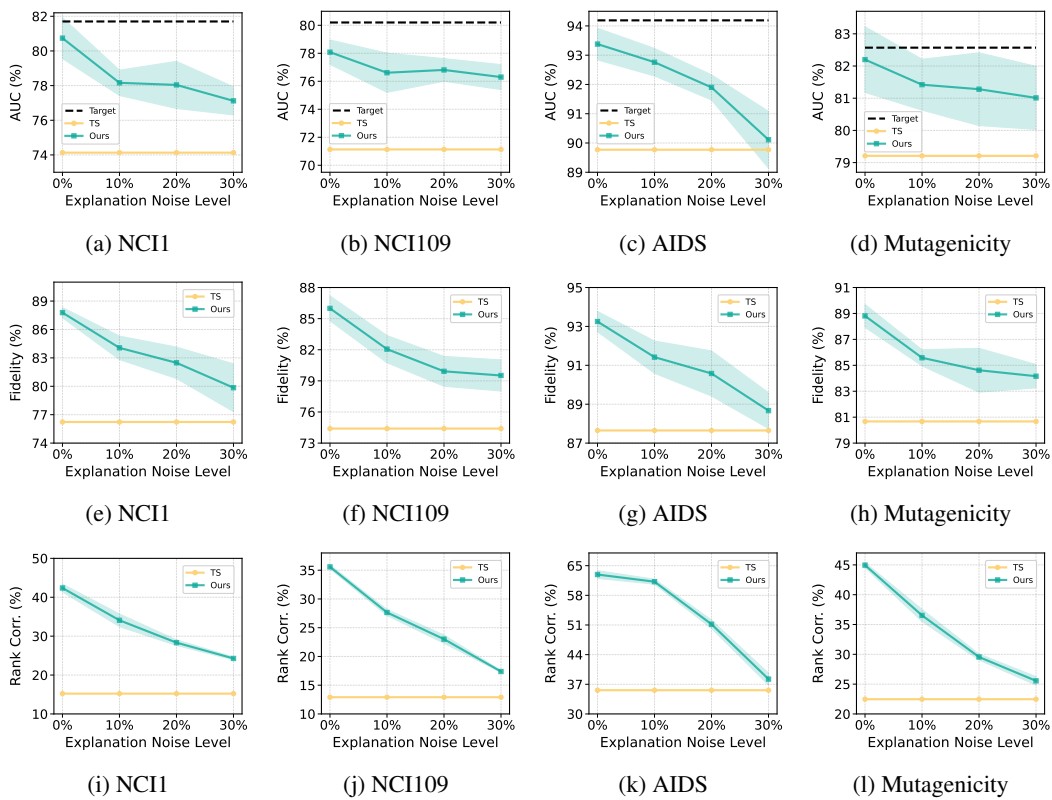

Figure 9: Performance under different explanation noise levels. Top row: AUC (%), Middle row: Fidelity (%), Bottom row: Explanation Rank Correlation (%). Each plot shows our method's performance degradation as noise increases from 0% to 30%, compared with TS baseline and target performance (AUC only).

**Complete Results for Binary Explanations** Figure 10 shows detailed performance comparison between continuous and binary explanation formats across all metrics. The binary explanation setting considers scenarios where attackers can only obtain binary information distinguishing important subgraphs from irrelevant parts. Despite this significant information reduction, our method consistently outperforms both the TS baseline and the binary explanation variant. The performance gap between our full method and the binary explanation variant varies across metrics, with explanation rank correlation showing the most pronounced reduction. This aligns with expectations that coarser explanation granularity limits the effectiveness of our rank-based alignment. Nevertheless, the binary explanation variant achieves notable improvements over the TS baseline, demonstrating the framework's adaptability to different explanation formats. The results suggest that even simplified explanation information provides valuable guidance for model extraction compared to pure prediction-based approaches.

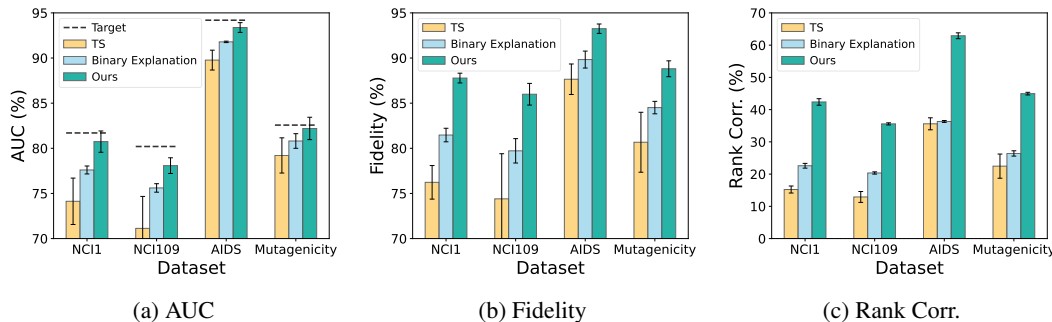

(a) AUC      (b) Fidelity      (c) Rank Corr.

Figure 10: Performance comparison with binary explanation format across datasets. Each subplot shows the performance of TS baseline, binary explanation variant, and our method on different datasets for AUC, fidelity, and explanation rank correlation metrics.

## G.5 ADDITIONAL RESULTS ON THE IMPACTS OF TARGET AND SURROGATE MODEL ARCHITECTURE

Figure 11 shows comprehensive performance gains of our explanation-guided approach over the teacher-student baseline across three metrics: AUC, fidelity, and explanation rank correlation.

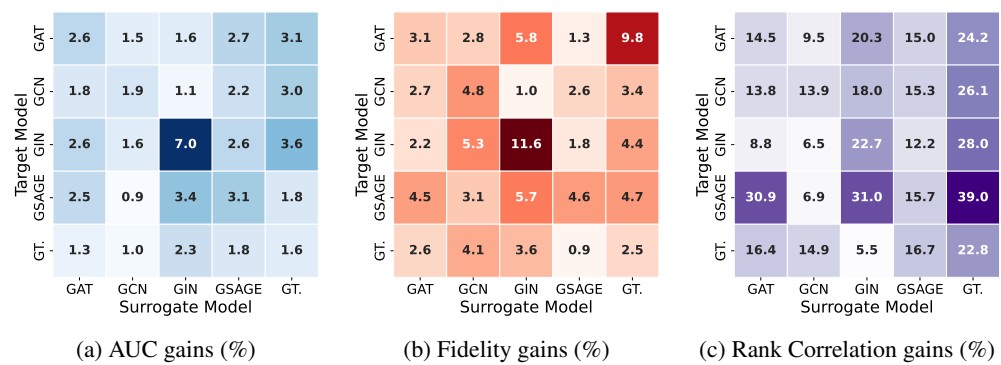

(a) AUC gains (%)      (b) Fidelity gains (%)      (c) Rank Correlation gains (%)

Figure 11: Performance improvements over teacher-student baseline with different model architectures on NCI109.

## G.6 ADDITIONAL RESULTS FOR DIFFERENT EXPLANATION MECHANISMS

Tables 7, 8, and 9 present additional results for different explanation mechanisms on NCI1, AIDS, and Mutagenicity datasets, respectively.

Table 7: Performance gains of different explanation mechanisms on NCI1

| Explainer | AUC (%) | Fidelity (%) | Rank Corr. (%) |
|---|---|---|---|
| Graph-CAM | $80.74 \pm 1.18$ ↑6.61 | $87.79 \pm 0.54$ ↑11.56 | $42.39 \pm 1.00$ ↑27.27 |
| Grad | $76.73 \pm 0.88$ ↑2.60 | $78.18 \pm 1.27$ ↑1.95 | $47.02 \pm 1.10$ ↑44.80 |
| Grad-CAM | $80.39 \pm 0.70$ ↑6.26 | $86.52 \pm 1.08$ ↑10.29 | $42.11 \pm 0.67$ ↑27.12 |
| GNNExpl. | $76.74 \pm 0.95$ ↑2.61 | $78.52 \pm 1.33$ ↑2.29 | $10.16 \pm 1.00$ ↑9.69 |
| PGExpl. | $76.60 \pm 1.89$ ↑2.47 | $78.32 \pm 0.81$ ↑2.09 | $69.10 \pm 0.78$ ↑69.00 |

Table 8: Performance gains of different explanation mechanisms on AIDS

| Explainer | AUC (%) | Fidelity (%) | Rank Corr. (%) |
|---|---|---|---|
| Graph-CAM | $93.38 \pm 0.54$ ↑3.61 | $93.25 \pm 0.52$ ↑5.60 | $62.91 \pm 0.91$ ↑27.31 |
| Grad | $91.25 \pm 1.54$ ↑1.48 | $88.40 \pm 1.71$ ↑0.75 | $56.18 \pm 0.42$ ↑40.93 |
| Grad-CAM | $93.42 \pm 0.60$ ↑3.65 | $92.15 \pm 1.31$ ↑4.50 | $61.72 \pm 1.23$ ↑28.22 |
| GNNExpl. | $90.66 \pm 1.40$ ↑0.89 | $86.50 \pm 1.39$ ↓1.15 | $10.78 \pm 0.98$ ↑10.17 |
| PGExpl. | $88.63 \pm 1.45$ ↓1.14 | $86.10 \pm 0.98$ ↓1.55 | $49.37 \pm 0.44$ ↑69.67 |

Table 9: Performance gains of different explanation mechanisms on Mutagenicity

| Explainer | AUC (%) | Fidelity (%) | Rank Corr. (%) |
|---|---|---|---|
| Graph-CAM | $82.20 \pm 1.24$ ↑2.99 | $88.81 \pm 0.88$ ↑8.14 | $44.96 \pm 0.41$ ↑22.49 |
| Grad | $79.80 \pm 1.12$ ↑0.59 | $81.20 \pm 1.48$ ↑0.53 | $65.16 \pm 0.33$ ↑53.02 |
| Grad-CAM | $80.15 \pm 0.66$ ↑0.94 | $81.73 \pm 1.04$ ↑1.06 | $22.51 \pm 1.26$ ↑19.83 |
| GNNExpl. | $81.45 \pm 2.04$ ↑2.24 | $82.38 \pm 0.50$ ↑1.71 | $28.02 \pm 0.67$ ↑26.60 |
| PGExpl. | $81.56 \pm 1.10$ ↑2.35 | $82.58 \pm 1.35$ ↑1.91 | $58.75 \pm 2.63$ ↑69.11 |

## G.7 COMPLETE ABLATION STUDY RESULTS

Figure 12 presents the complete ablation study results across all performance metrics.

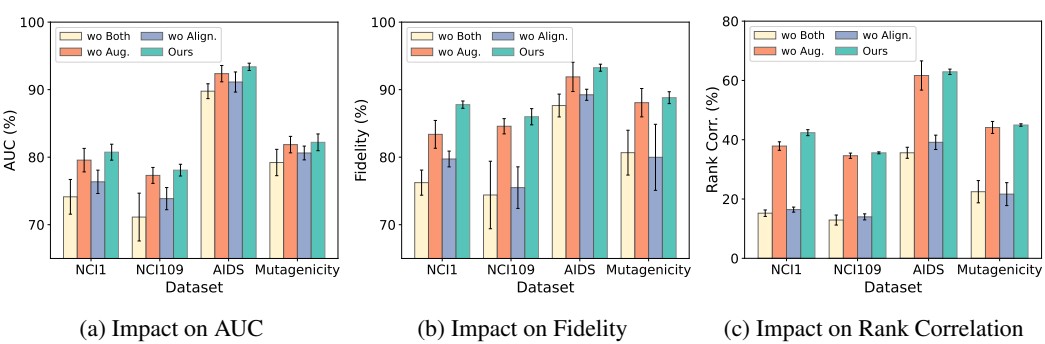

(a) Impact on AUC        (b) Impact on Fidelity        (c) Impact on Rank Correlation

Figure 12: Ablation study results across four datasets showing the impact of removing data augmentation and explanation alignment components from our full framework.

## G.8 HYPERPARAMETER ANALYSIS

We conduct comprehensive parameter sensitivity analysis focusing on two key hyperparameters: the data augmentation ratio $\alpha$ (controlling the proportion of augmented samples) and the alignment loss coefficient $\lambda$ (balancing prediction matching and explanation alignment). We vary $\alpha$ from 0.1 to 0.5 and $\lambda$ from 0.01 to 100, conducting experiments on the NCI109 dataset with settings consistent with Section 5.2. As shown in Figure 13b, fidelity consistently improves as alignment coefficient $\lambda$ increases, while augmentation ratio $\alpha$ achieves optimal results between 0.1 and 0.3. Figure 13a reveals that neither excessively small nor large augmentation ratio $\alpha$ yields optimal AUC performance, suggesting that a moderate amount of data augmentation is most beneficial for model stealing. Regarding the alignment of explanations illustrated in Figure 13c, while the data augmentation ratio $\alpha$ demonstrates relatively consistent impact, the alignment coefficient $\lambda$ plays a crucial role in capturing the target model's decision logic. As $\lambda$ increases from 0.01 to 10, we observe significant improvement in the consistency of node importance rankings between the surrogate and target models, indicating enhanced capture of the target model's decision logic. However, when $\lambda$ exceeds a certain threshold, the improvement becomes marginal and may even compromise the surrogate model's ability to match the target model's predictions.

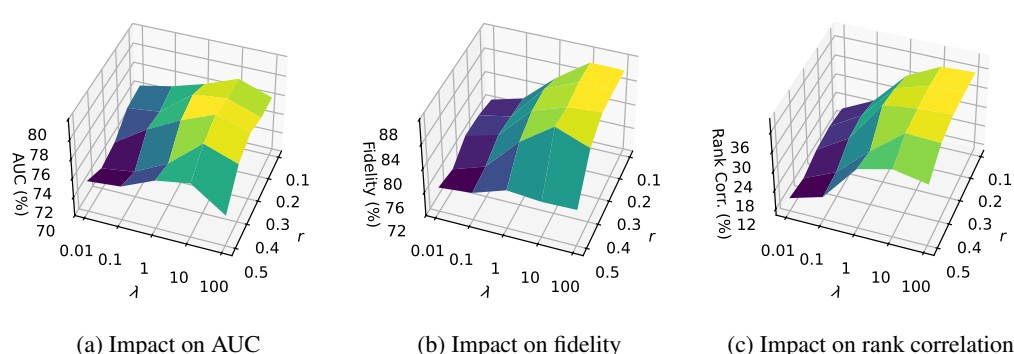

(a) Impact on AUC        (b) Impact on fidelity        (c) Impact on rank correlation

Figure 13: Sensitivity analysis of data augmentation ratio $\alpha$ and alignment loss coefficient $\lambda$.

## G.9 TRAINING TIME

We followed the experimental settings described in Section 5.2 to measure the training time of the surrogate model, conducting runtime evaluations comparing EGSteal to other baseline methods across multiple datasets. As shown in the Table 10, although EGSteal requires more training time than some simpler baselines, its computational overhead is not the highest and remains reasonable, which is acceptable considering the performance gains achieved.

Table 10: Training time comparison (in seconds)

| Dataset | TS | MEA-GNN | GNNStealing | EfficientGNN | MRME | DET | STEALGNN | Ours |
|---|---|---|---|---|---|---|---|---|
| NCI1 | 25.87 | 26.70 | 48.81 | 55.66 | 313.48 | 60.50 | 82.43 | 109.18 |
| NCI109 | 26.31 | 27.25 | 51.82 | 59.04 | 330.24 | 65.03 | 85.25 | 109.42 |
| AIDS | 10.37 | 9.40 | 21.58 | 36.79 | 180.18 | 26.85 | 45.66 | 54.58 |
| Mutagenicity | 28.33 | 24.84 | 55.42 | 79.08 | 308.64 | 64.75 | 81.09 | 111.75 |
| HIV | 75.97 | 80.64 | 105.78 | 235.40 | 1072.27 | 160.05 | 213.56 | 319.08 |
| Tox21 | 25.89 | 35.02 | 45.85 | 45.38 | 574.38 | 66.15 | 89.72 | 162.77 |
| BACE | 11.85 | 7.62 | 19.71 | 30.29 | 141.59 | 18.66 | 29.87 | 41.72 |
| PubMed | 29.04 | 26.26 | 63.53 | 73.86 | 413.43 | 106.95 | 147.21 | 152.13 |
| ogb-arxiv | 60.72 | 59.17 | 137.27 | 144.82 | 1028.85 | 167.99 | 220.02 | 435.28 |

## G.10 CASE STUDY

We present visual analysis of molecular substructures identified as important by both target and surrogate models to demonstrate the effectiveness of our explanation alignment mechanism. Figure 14 shows examples comparing the explanations generated by the target model and our surrogate model across different molecular structures, revealing highly similar substructure patterns between the target (blue) and surrogate (red) models. The consistent overlap between blue highlights (target model explanations) and red highlights (surrogate model explanations) across these examples confirms that our surrogate model effectively captures the target model's decision-making patterns and validates that our framework successfully replicates not only the predictive behavior but also the underlying reasoning process of the target model.

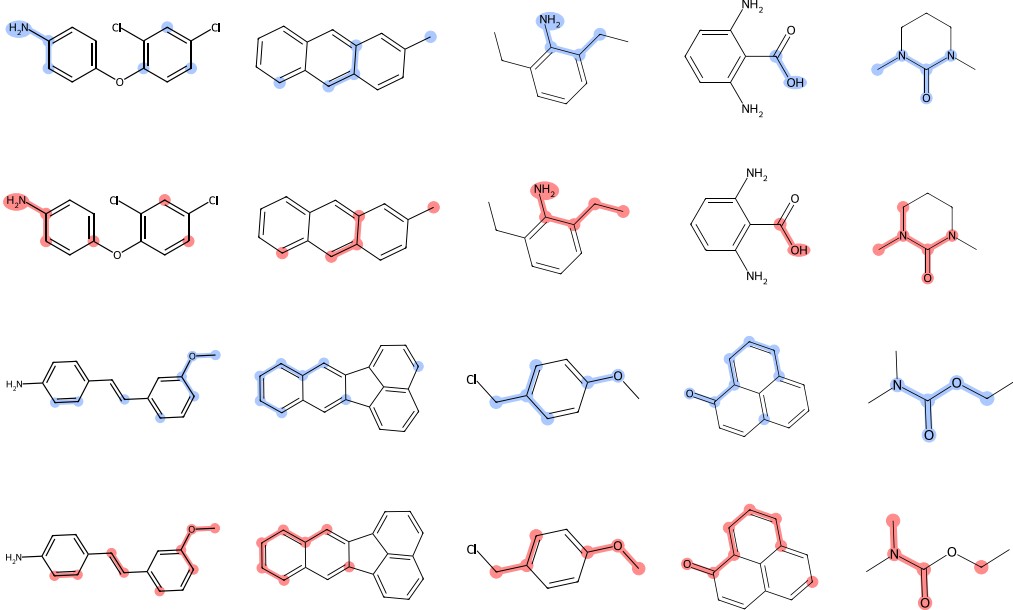

Figure 14: Visualization of molecular substructures identified as important by target and surrogate models. Blue highlights indicate target model explanations, while red highlights show surrogate model explanations.