# OpenReview forum: "How Explanations Leak the Decision Logic: Stealing Graph Neural Networks via Explanation Alignment"
_ICLR.cc/2026/Conference — Submitted to ICLR 2026_

### Official Review · Reviewer_njmV · 2025-10-27

**Soundness:** 2
**Presentation:** 3
**Contribution:** 3
**Rating:** 4
**Confidence:** 4

**Summary:**

Since GNNs are widely used in many high-stake domains, the explanations on how GNN make predictions are becoming important. Existing GNN explanations can be obatined either by post-hoc explanations or self-explainable GNNs. While being useful to increase the transparency of decision making, it also increases the risk that the GNNs can be attacked (with the help of these explanations). This paper investigates the so-called model stealing attacks (namely replicating the prediction behaviour of the GNNs) with the help of GNN explanations. To achieve this, they assume that a graph can be devided into two groups: explanation subgraphs and style subgraphs. They further (strongly) assume that the explanation subgraph fully determinate the prediction while the other subgraphs have no impact on the GNN prediction. On this basis, they design a mimic model that not only aligns the predicted label but also the provided explanation subgraph. To train such a model, they augment the training pool by perturbing the style subgraphs without querying the target GNN model too many times. They perform many empirical studies to show the effectiveness and superiority of their method. However, I have many concerns that need more empirical studies, without which the soundness of this paper can be largely weakened.

**Strengths:**

1. This paper is well organized;
2. Many statements are supported with experiments;
3. They tackle a novel problem;

**Weaknesses:**

The authors assume that the provided explanation subgraph fully determinate the GNN prediction. However, most post-hoc explanations may not reflect the actual decision logic [1]; besides, even self-explainable GNNs may provide explanations that are different from the underlying subgraphs that actually drived the predictions [2][3]. This issue is called unfaithful explanations, which have been widely recognized. Let me make this comment more actionable:

(1) how could you guarantee (or measure) that the provided explanations actually reflect the decision logic, especially for post-hoc explanations? and under a black-box setting?

(2) if you could not guarantee (or measure) this, what is the impact of the faithfulness of explanations on your method? Quantitaive and/or qualitative analyses are necessary;

(3) what if there are multiple explanation subgraphs that align with the decision logic, but the explainer only provides a single one? You will treat the others as style subgraphs that do not impact the prediction? What will be the impact on your methodology?

(4) What if the GNN predictor is a weak predictor? In other words, if the clasisfication accuracy is low (for example 0.4), the GNN classifier may not follow the true decision logic (if they follow the true decision logic, the accuracy should be high). Will your stealing model be able to align with the GNN preidction as well as the explanation?

## References
[1] Faithful and Consistent Graph Neural Network Explanations with Rationale Alignment

[2] How Faithful are Self-Explainable GNNs?

[3]  GNN Explanations that do not Explain and How to find Them

PS: I am willing to increase my ratings if my concerns are (partially) addressed. (I may lower my ratings if they are completely ignored.)

**Questions:**

Please see the weak points

---

> ### Author Response · Authors · 2025-11-23
>
> > Response to W1&W2:
>
> "How could you guarantee (or measure) that the provided explanations actually reflect the decision logic, especially for post-hoc explanations? And under a black-box setting?"**
> **"If you could not guarantee (or measure) this, what is the impact of the faithfulness of explanations on your method? Quantitative and/or qualitative analyses are necessary."
>
> We would like to clarify that our EGSteal does **NOT** assume that the provided explanations fully reflect the model's decision logic. Instead, EGSteal aims to demonstrate the risk of model stealing when GNN predictions are accompanied by various explanations, including gradient-based explanations, optimization-based post-hoc explanations, and even noisy explanations.
>
> We agree with the reviewer that existing explanation methods may not perfectly capture the model's decision logic for all samples. However, since these imperfect explanations are currently prevalent in practice and can still partially reflect the underlying decision logic, investigating their associated risks of model decision leakage remains valuable and necessary.
>
> Next, we analyze the impact of explanation faithfulness on our method from two complementary perspectives: (1) evaluating the faithfulness of explanations from different explainers and their effectiveness in our framework, and (2) systematically degrading explanation quality to understand the robustness of our method.
>
> ### 1. Faithfulness of explanations from different explainers
>
> To measure explanation faithfulness without ground-truth annotations, we follow established practices in GNN explainability research [1][2][3] by measuring whether the important subgraph identified by an explanation preserves the target model's prediction on the original graph:
>
> $$\text{Faithfulness}=\frac{1}{|\mathcal{D}|}\sum_{\mathcal{G}\in\mathcal{D}}\mathbb{1}[y\_{\theta}(\mathcal{G}\_E)=y\_{\theta}(\mathcal{G})],$$
>
> where $\mathcal{G}_E$ is the subgraph containing top-ranked nodes according to the explanation. Higher faithfulness indicates that the explanation better captures the decision-relevant structures.
>
> We evaluated the faithfulness of five different explainers on the NCI1 dataset:
>
> |Explainer|Faithfulness (%)|AUC (%)|Fidelity (%)|Rank Corr. (%)|
> |---|---|---|---|---|
> |Graph-CAM|81.91|80.74±1.18|87.78±0.54|42.38±1.01|
> |Grad|80.26|78.93±1.24|84.32±1.18|48.15±1.35|
> |Grad-CAM|81.05|79.85±1.09|86.51±0.97|40.27±0.89|
> |GNNExplainer|75.97|77.28±1.47|81.94±1.53|25.83±1.12|
> |PGExplainer|73.02|77.65±1.31|82.17±1.68|58.42±1.27|
> |TS baseline|—|74.13±2.57|76.23±1.86|15.22±1.09|
>
> **Key observations:**
>
> - Faithfulness ranges from 73% to 82% across all explainers on NCI1. Despite these variations, current explanation methods provide signals about the target model's decision logic that can be effectively exploited for model stealing, as evidenced by consistent improvements over the TS baseline across all explainers.
> - Our framework achieves substantial improvements over the baseline regardless of the specific explainer mechanism, demonstrating that the explanation alignment mechanism can leverage diverse explanation formats and that EGSteal effectiveness is robust across different explainer types.
>
> ### 2. Impact of degraded explanation faithfulness through noise injection
>
> As discussed in Section 5.3 (Robustness Analysis) and Appendix G.4, we systematically degraded explanation quality by randomly shuffling a fraction of the target model's explanation outputs to disrupt the original node importance ranking. This simulates scenarios where explanations are unreliable or corrupted. We measured the faithfulness at each noise level to quantify the degradation. Results on NCI1 are shown below:
>
> |Noise Level|Faithfulness (%)|AUC (%)|Fidelity (%)|Rank Corr. (%)|
> |---|---|---|---|---|
> |0%|81.91|80.74±1.18|87.78±0.54|42.38±1.01|
> |10%|78.24|78.16±0.73|84.06±1.24|34.08±1.62|
> |20%|74.02|78.04±1.36|82.48±1.66|28.33±0.72|
> |30%|71.81|77.12±0.81|79.85±2.52|24.27±0.36|
> |TS baseline|—|74.13±2.57|76.23±1.86|15.22±1.09|
>
> We make the following observations:
> - Performance systematically decreases as explanation quality is degraded. This demonstrates that explanations contain useful information about the target model's decision behavior. If explanations were uninformative, noise injection would not affect performance.
> - Even with 30% noise, our method still outperforms pure prediction-based stealing (TS baseline), demonstrating that the explanation alignment mechanism of our EGSteal can effectively leverage the partial signals present to capture the target model's reasoning patterns.
>
> [1] Explainability in graph neural networks: A taxonomic survey
> [2] On Explainability of Graph Neural Networks via Subgraph Explorations
> [3] Parameterized Explainer for Graph Neural Networks

---

> ### Author Response · Authors · 2025-11-23
>
> > Response to W3: Concerns about scenarios with multiple explanation subgraphs
>
> The explainer we employ produces continuous importance scores ($E_v\in[0,1]$) for each node rather than discrete classifications of subgraphs. Our method (Eq. 8) uses ranking-based selection to preserve top-ranked nodes through soft selection rather than hard partitioning.
>
> If multiple substructures contribute to the prediction, the nodes of all these structures will receive high importance scores. For example, in a molecular graph where both a benzene ring and a carboxyl group are important, nodes from both structures will receive high importance scores and be retained in the explanation subgraph $\mathcal{G}_E$, rather than being misclassified as style subgraphs.
>
>
>
> > Response to W4: Concerns about stealing weak predictors with low classification accuracy
>
> This concern conflates two concepts: the model's learned decision logic (what the model actually relies on) and the ground-truth causal logic (the true mechanisms). Model stealing aims to replicate the former, not recover the latter.
>
> For a weak classifier, successful stealing should produce a surrogate with similar accuracy and high prediction fidelity, replicating the target model's behavior, including its weaknesses.
>
> To empirically validate this, we created underfitted weak classifiers by reducing training epochs:
>
> |Dataset|Metric|Target|TS|Ours|
> |---|---|---|---|---|
> |NCI1|AUC|73.05%|68.76±1.26%|72.27±1.29%|
> ||Fidelity|–|88.76±0.98%|91.63±1.29%|
> |NCI109|AUC|72.76%|69.88±1.27%|71.44±0.91%|
> ||Fidelity|–|91.83±0.87%|96.07±1.17%|
> |AIDS|AUC|88.41%|82.57±2.57%|87.45±1.03%|
> ||Fidelity|–|89.25±2.39%|93.00±2.92%|
> |Mutagenicity|AUC|78.70%|75.74±1.26%|78.25±0.83%|
> ||Fidelity|–|87.66±1.21%|91.95±1.38%|
>
> Our method achieves AUC close to the weak targets while maintaining high fidelity (89–96%), demonstrating successful replication of decision boundaries, including their limitations. This validates that model-stealing effectiveness does not depend on the target model's absolute performance level.

---

> > ### Comment · Reviewer_njmV · 2025-11-25
> >
> > Dear Authors,
> >
> > Thank you for yor reply. I am satisfied with the rebuttal (especially regarding weak points 1 and 2). Therefore, I decide to increase my overall rating from 4 to 6. I hope you can incoporate the responses to weak point 1, 3, 4 in the revised version (even if this paper is not accepted by ICLR this time). (Pherhaps I will also be the reviewer again for this paper if you resubmit it to ICML/NIPS or other good venues...)

---

> > > ### Author Response · Authors · 2025-11-25
> > >
> > > Dear Reviewer njmv,
> > >
> > > Thank you for acknowledging that our clarifications have addressed your questions. We will incorporate the updated content into the next version of our manuscript.
> > >
> > > Best regards,
> > > Authors

---

### Official Review · Reviewer_1dwp · 2025-11-01

**Soundness:** 2
**Presentation:** 3
**Contribution:** 2
**Rating:** 4
**Confidence:** 4

**Summary:**

This paper propose a GNN model extraction attack method specifically targeting models with explanation subgraph provided.  The paper utilizes the causal relation between explanation in graph and label outcome to construct intervened graphs as augmented data. The author further design a training loss for surrogate model with explanation output to ensure both it replicates the classification result and interpretable mechanism.

**Strengths:**

**1.** The idea of not only aligning mode classification but also aligning model explanation outcome is interesting. By requiring the surrogate model to have similar importance preference with target model, it may potentially make their inner mechanisms to become closer, achieving a better replication.

**2.** The presentation of the paper is clean and sound. The notation in the paper is used clearly and formulations are also expressed tidily. The figures for illustration are easy to understand.

**3.** The experimental results provided are reliable. There are many details of the algorithm design and hyper-parameters setting are discussed in the appendix, and the code is provided through the anonymous link. Attack performance under different attack capacity and architectures are also fully test, making the experimental results reliable.

**Weaknesses:**

**1.** The proposed intervened data as augmented data may introduce "incorrectness". In the causal analysis the author assumed the explanation subgraph $G_{E}$ is the part that decides to the classification,  while the style graph has little impact. So the author arbitrarily perturb the style graph while holding explanation subgraph to be the same to generate the intervened graph, while given the original label and same explanation subgraphs. However, this may be incorrect in some cases. For example, if we hope to indicate a benzene structure in a molecular (6 edges in a ring), the generated intervened graph may linking edges between them, and make the graph indeed no longer contain clean benzene and GT label changed, but the augmented data is still labeled as "contain", so incorrectness happens; the intervened model may also be created with benzene in $G_{S}$ and lead the real $G_{E}$ change, while the augmented data still holds the original one. In all, the assumption on Eq.(3) lacks of rationality since sometimes GNN not classify a specific subgraph region but a structure pattern, which should vary when $G_{S}$ change.

**2.** The chosen datasets all contain a lower quantity of average graph size within the dataset and lacks ground-truth explanations. The proposed intervened graphs hold a relatively large perturbation space, and when the graph scale is large, it would require much more samples to satisfy the Eq.(3) and Eq.(4). However, the chosen datasets in the experiments are relatively small on every single graph, which omits this potential problem. Besides, the chosen datasets do not contain a naturally existent groundtruth explanation, which makes it hard to known if the explained results $G_{E}$ really have the equality to hold Eq.(3) and (4).

**Questions:**

**1.**  As stated in appendix, the proposed loss term requires a $TQ|V|^{2}$ complexity to calculate all pairs of nodes' importance, which is a relatively high complexity. Be So I wonder if there's runtime record for conducting an extraction attack.

**2.** Following the complexity problem, it seems that the main burden comes from the loss term on aligning explanation ranking on the surrogate model and target model. So how about remove this term? Would this largely decrease the model's performance?

**3.** Is the surrogate model's explainer required to be the same with the target model? If not, would this architecture shift introduce additional error on replication?(ablation study suggested) If yes, this setting seems to lose practicability since the target model's explainer may not be known to the attacker, e.g., architecture or inner parameters.

I suggest the author to at least conduct ablation study on the **questions 2&3**, because it leads to the necessity and rationality of designing the alignment loss term, which is currently unclear on its utility compared with its high complexity. If these two questions are properly addressed I would like to **raise the score to at least positive**.

---

> ### Author Response · Authors · 2025-11-23
>
> > Response to W1: Potential Incorrectness in Augmented Data
>
> We acknowledge that data augmentation inherently introduces some noise, which is an unavoidable trade-off in augmentation strategies. However, the noise introduced by our augmentation strategy is not severe, precisely because we only perform random perturbations on style subgraphs while preserving the causal explanation subgraphs. Our method operates on node-level importance rankings: if a structural pattern is causally important, its constituent nodes receive high importance scores and are preserved in $\mathcal{G}_E$.
>
> To evaluate the quality of augmented samples, we fed both original and augmented graphs to the target model and measured the prediction consistency between them:
>
> |Dataset|Prediction Fidelity (%)|
> |---|---|
> |NCI1|95.92|
> |NCI109|95.40|
> |AIDS|96.25|
> |Mutagenicity|98.16|
>
> Results show greater than 95% prediction fidelity across all datasets, indicating that augmented graphs preserve the target model's predictions. This empirically verifies that the "incorrectness" introduced by the intervention is negligible.
>
>
>
> > Response to W2: Limited Dataset Scale and Lack of Ground-Truth Explanations
>
> We would like to clarify that beyond graph classification on molecular datasets, our evaluation includes node classification on large-scale graphs, including PubMed and ogbn-arxiv, as detailed in Section 5.4. These experiments demonstrate that our framework effectively handles large graphs. The molecular graph datasets we employed are commonly used in GNN explainability research and represent domains where explainability is critical. While our datasets lack annotated ground-truth explanations, this is a common constraint in explainability research and does not undermine our approach.
>
> Our objective is to leverage the target model's explanation outputs to steal its decision logic, specifically which subgraph parts influence the model's final prediction, allowing the surrogate model to better fit the target model's behavior through the explanation alignment mechanism. Even if the target model's explanation outputs are not perfect ground truth, this does not prevent us from capturing its decision logic.

---

> ### Author Response · Authors · 2025-11-23
>
> > Response to Q1: Runtime Complexity and Training Time
>
> We have provided comprehensive complexity analysis in Appendix D and runtime measurements in Appendix G.9. For convenience, we reproduce the training time (in seconds) comparison below:
>
> |Dataset|TS|MEA-GNN|GNNStealing|EfficientGNN|MRME|DET|STEALGNN|Ours|
> |---|---|---|---|---|---|---|---|---|
> |NCI1|25.87|26.70|48.81|55.66|313.48|60.50|82.43|109.18|
> |NCI109|26.31|27.25|51.82|59.04|330.24|65.03|85.25|109.42|
>
> Due to space constraints, complete results for all 9 datasets are available in Appendix G.9.
>
> Although EGSteal requires more training time than some simpler baselines, its computational overhead is not the highest and remains reasonable, which is acceptable considering the performance gains achieved.
>
>
>
> > Response to Q2: Ablation Study on Removing Alignment Loss
>
> We clarify that this has been analyzed in Section 5.5 and Appendix G.7. We evaluated four variants (w/o Both, w/o Augmentation, w/o Alignment, Ours) with results shown below:
>
> |Dataset|Metric|w/o Both|w/o Aug.|w/o Align.|Ours|
> |---|---|---|---|---|---|
> |NCI1|AUC|74.13±2.57|79.56±1.74|76.35±1.73|80.74±1.18|
> ||Fidelity|76.23±1.86|83.38±2.06|79.73±1.17|87.78±0.54|
> ||Explanation Rank Corr.|15.22±1.09|37.86±1.44|16.43±0.85|42.38±1.01|
> |NCI109|AUC|71.13±3.54|77.30±1.18|73.86±1.64|78.08±0.87|
> ||Fidelity|74.40±5.00|84.58±1.13|75.49±3.08|85.99±1.20|
> ||Explanation Rank Corr.|12.91±1.69|34.60±0.89|13.98±1.02|35.58±0.33|
>
> Due to space constraints, complete results for all 4 datasets are available in Appendix G.7.
>
> Removing the alignment loss causes performance degradation, particularly with explanation rank correlation dropping to near baseline levels across all datasets. This demonstrates the importance and effectiveness of the alignment loss for capturing the target model's explanation patterns.
>
>
>
> > Response to Q3: Requirement for Explainer Consistency
>
> Our method does not require the surrogate model's explainer to match the target model's explainer. This is empirically demonstrated in Section 5.4 (Flexibility to Explainers) and Appendix G.6. For convenience, we reproduce the results on NCI109 showing performance across five different target explainers:
>
> |Explainer|AUC (%)|Fidelity (%)|Explanation Rank Corr. (%)|
> |---|---|---|---|
> |Graph-CAM|78.08±0.87 (↑6.95)|85.99±1.20 (↑11.59)|35.58±0.33 (↑22.67)|
> |Grad|75.38±1.08 (↑4.25)|78.50±0.94 (↑4.10)|52.25±1.01 (↑48.40)|
> |Grad-CAM|77.38±1.10 (↑6.25)|85.43±2.30 (↑11.03)|36.60±0.79 (↑23.68)|
> |GNNExpl.|75.12±1.59 (↑3.99)|77.75±1.63 (↑3.35)|20.25±0.62 (↑22.76)|
> |PGExpl.|75.48±1.28 (↑4.35)|77.99±2.31 (↑3.59)|64.07±0.79 (↑65.03)|
>
> Numbers in parentheses indicate gains over the TS baseline.
>
> When the target model uses different explainers, our method consistently achieves substantial improvements over the TS baseline across all metrics. The ranking-based alignment mechanism operates on node importance rankings, which allows the surrogate model to learn from different explanation representations. Furthermore, as shown in Section 5.4, when the target model and surrogate model use different GNN backbones, performance improvements remain consistent. Our method does not assume that the target model and surrogate model use the same architecture.

---

### Official Review · Reviewer_kv62 · 2025-11-01

**Soundness:** 2
**Presentation:** 3
**Contribution:** 2
**Rating:** 2
**Confidence:** 4

**Summary:**

This paper shows that model explanations can leak decision logic and enable model stealing. The authors propose EGSteal, an attack framework that combines explanation alignment and data augmentation to replicate a target model’s predictions and reasoning under limited queries.

**Strengths:**

- The paper is clearly written and well structured, making it easy to follow the authors' ideas and understand the main contributions.
- The figures are well designed and intuitive, effectively illustrating how the proposed method works and helping readers grasp the key concepts at a glance.

**Weaknesses:**

- The paper claims that GNN models are deployed online to protect intellectual property in critical applications (e.g., drug screening). However, in realistic IP-protection scenarios, online ML services are unlikely to provide model explanations, since explanations can reveal internal model logic and sensitive knowledge.

- Only explanations that rely on internal model access (e.g., gradient- or attention-based, or self-interpretable models) provided by an online ML service are useful; for post-hoc explainers such as GNNExplainer or PGExplainer, attackers could obtain explanations locally. Suppose the target model relies on a post-hoc explainer such as PGExplainer. In that case, it is questionable whether the proposed method is stealing the GNN model itself or rather the explainer.  Even if explanations were accessible, different explanation methods often yield inconsistent results, so aligning a surrogate model based on Graph-CAM explanations without knowing the target’s explanation mechanism is weird.

- The proposed data augmentation is motivated by the assumption that real-world APIs limit the number of allowed queries. However, no empirical evidence or real-world example is provided to support this claim. The authors should show that such query constraints exist in real-world services.

- The theoretical assumption that $G_S$ is prediction-irrelevant is inconsistent with the implementation, where $G_S$ is determined by a rank-based threshold in Eq. (8). This creates two issues: (1) If $G_S$ is truly irrelevant, node scores should be 0 (or close to 0), rather than merely ranked low, since even low-ranked nodes could still have non-negligible importance; (2) The manual hyperparameter $\alpha$ critically affects the construction of $G_E$ and $G_S$, as well as the rationality of data augmentation.

- If the target model is truly a black box, the attacker should not know which explanation method the target uses (e.g., post-hoc vs. self-interpretable). However, most main experiments assume Graph-CAM explanations for the target model, which is overly idealized.

- In the cross-dataset setting, the evaluation focuses on showing that the proposed method outperforms TS in performance metrics (Lines 396-398). However, for an attack method, the key objective should be whether the surrogate model truly replicates the target model’s behavior. The results on the AIDS dataset, where the AUC is below 60%, suggest that the proposed method may fail to effectively steal the target model when the in-distribution assumption does not hold. This raises concerns about the validity of the method under realistic scenarios.

**Questions:**

- The paper focuses on node-level explanations, but in graph applications, structure-level (i.e., edge-level) explanations are more common. Could the proposed method be directly extended to edge-level explanations? I suspect that, due to the specific design of the ranking-based loss, the proposed method might not generalize well to edge-level explanations.
- The paper presents an experiment showing the robustness of the proposed method to noisy explanations. Could the authors clarify: (1) The motivation for this experiment: In what practical scenarios would the explanations become noisy? (2) Why is the proposed method robust to such noise? Is there any specific mechanism or design choice that contributes to this robustness?
- In the experiments, the query budget is set to at least 10% of the training dataset. This means the attacker can issue hundreds or even thousands of queries. Is it motivated by any real-world scenario (e.g., are there existing systems that limit users to a few hundred queries)?


**I found several aspects of the paper unconvincing as currently presented, which leads me to recommend rejection. That said, I may have misunderstood some points, and I would appreciate further clarification from the authors during the rebuttal phase. I apologize if any of my remarks are based on a misunderstanding, and I thank the authors in advance for clarifying these points.**

---

> ### Author Response · Authors · 2025-11-23
>
> > Response to W1: Concerns about the realism of providing explanations in IP-protection scenarios
>
> Currently, deployed models providing explainable analysis have not yet been widely adopted in production environments. However, this situation is rapidly changing as explainability transitions from optional to mandatory due to regulatory requirements in critical domains.
>
> - The EU AI Act requires high-risk AI systems, including those in drug discovery and healthcare, to provide sufficient transparency enabling users to interpret system outputs.
> - The US Food and Drug Administration (FDA) emphasizes that medical AI devices should provide clear explanations for their decisions to ensure accountability and build trust.
> - The European GDPR's right to explanation requires providing meaningful information about the logic involved in automated decision-making.
>
> These regulations create scenarios where service providers must balance IP protection with legal compliance, precisely the tension our work addresses.
>
> While widespread deployment of explanation-providing services is still emerging, this strengthens our work's significance. We identify a critical security vulnerability before explainability becomes ubiquitous, providing insights for developing defense mechanisms. As regulatory requirements take full effect, organizations will face the challenge of providing compliant explanations while protecting IP. Our work provides the first systematic analysis of this risk, demonstrating that explanation exposure creates severe model stealing vulnerabilities.
>
>
>
> > Response to W2: Concerns about post-hoc explainers and explanation alignment
>
> This concern appears to stem from overlooking key aspects of our problem formulation and methodology already detailed in the paper. We clarify three fundamental points below.
>
> First, regarding what is being stolen: As detailed in Section 3 (Preliminaries), our method aims to steal the GNN model itself, not the explainer. In fact, we test different explainer types for the target model in our experiments in Section 5.4 and achieve consistent improvements across various explainers (Table 4). This demonstrates the flexibility of our method, which does not require any assumptions about the target model's explainer type.
>
> Second, regarding explanation inconsistency across methods: As described in Section 4 (Methodology), we explicitly do not align raw explanation values precisely because different explainers produce outputs with varying scales and distributions. Instead, we align the relative ranking of node importance. As stated in the paper: "despite variations in magnitude, the relative ordering of node importance tends to remain stable and effectively reflects the underlying reasoning process." This ranking-based mechanism is robust to the specific explainer used, making our approach effective regardless of the target's explanation mechanism.
>
> Third, regarding the choice of Graph-CAM for the surrogate model: Section 4.2.1 explicitly explains why we adopt Graph-CAM as the surrogate's explainer: it efficiently obtains node importance scores without additional explainer training, and its gradient-based nature enables direct backpropagation for optimization. This is a technical design choice for the surrogate model's internal mechanism. The target model can employ any explanation method, and as demonstrated in our experiments in Section 5.4, our framework remains effective across different explainers used by the target model.
>
>
>
> > Response to W3 & Q3: Concerns about query budget assumptions
>
> Query budget constraints are realistic and well-motivated for several reasons:
>
> - **Commercial practice:** API query limits are standard in production systems for controlling operational costs and defending against malicious mass querying, including model extraction and denial-of-service attacks. For example, commercial services like Google's Gemini API and OpenAI's GPT models implement rate limits and usage quotas for API access.
> - **Financial constraints:** API access is typically not unlimited or free. Realistic model stealing attacks must account for the financial costs associated with querying commercial services, which naturally limits the number of queries attackers can afford.
> - **Research standards:** Query budget constraints are standard assumptions in model stealing research [1][2][3], reflecting realistic attacker constraints in practical attack scenarios. Our experimental setting aligns with these established practices and represents a realistic yet challenging attack scenario.
>
> [1] Model stealing attacks against inductive graph neural networks
> [2] Efficient model-stealing attacks against inductive graph neural networks
> [3] Model extraction attacks on graph neural networks: Taxonomy and realisation

---

> ### Author Response · Authors · 2025-11-23
>
> > Response to W4: Concerns about theoretical assumptions and implementation
>
> We use hyperparameter α to partition the causal subgraph from the style subgraph based on node importance rankings, which is a common practice in GNN explainability research. Explanation methods such as GNNExplainer [1] and PGExplainer [2] typically obtain soft masks for nodes or edges and select important subgraphs by applying a threshold. Theoretically, the causal assumption is that the style subgraph $G_S$ has negligible impact on predictions. In practice, importance scores in GNNs form continuous distributions without strict zeros, so "prediction-irrelevant" refers to negligible influence relative to the causal subgraph, which is consistent with how explanations work in real GNN models.
>
> We validated this through subgraph fidelity experiments comparing the target model's predictions on causal subgraphs $G_E$ versus style subgraphs $G_S$. Fidelity measures the prediction agreement:
>
>
> $$\text{Fidelity}=\frac{1}{|\mathcal{D}|}\sum_{\mathcal{G}\in\mathcal{D}}\mathbb{1}[y\_{\theta}(\mathcal{G}\_{\text{sub}})=y\_{\theta}(\mathcal{G})],$$
>
>
> where $\mathcal{G}_{\text{sub}}$ is the extracted subgraph (causal or style) and $\mathcal{G}$ is the original graph.
>
> |Dataset|$G_E$ Fidelity (%)|$G_S$ Fidelity (%)|
> |---|---|---|
> |AIDS|93.99|43.65|
> |NCI1|82.06|52.31|
> |NCI109|85.23|47.94|
> |Mutagenicity|92.97|44.01|
>
> Causal subgraphs maintain high prediction consistency with original predictions, while style subgraphs show near-random predictions. This demonstrates that our ranking-based partitioning effectively separates nodes with significant predictive impact from those with negligible influence, validating the theoretical foundation of our approach.
>
> [1] GNNExplainer: Generating Explanations for Graph Neural Networks
> [2] Parameterized Explainer for Graph Neural Network
>
>
>
> > Response to W5: "If the target model is truly a black box, the attacker should not know which explanation method the target uses"
>
> Our method does **NOT** assume knowledge of the target's explanation mechanism. In Section 5.4, we evaluated the attack across different explanation methods without any information of the explainer methods. All achieve substantial improvements over baselines, demonstrating that the method is robust to explainer choice and does not depend on any specific explanation mechanism.
>
>
>
> > Response to W6: Concerns about cross-dataset evaluation
>
> The cross-dataset experiments are designed to evaluate robustness under distribution shift, not to demonstrate perfect stealing in out-of-distribution scenarios. When shadow data distribution diverges from the target model's training distribution, performance degradation is expected and natural across all methods. The critical observation is that despite this distribution mismatch, our method maintains consistent advantage over the baseline, demonstrating the robustness and flexibility of our approach in challenging scenarios. Addressing severe distribution shift under out-of-distribution conditions is beyond the scope of our research; our contribution is demonstrating that explanation-based alignment provides consistent benefits even when attackers face distribution mismatch, compared to baseline methods, showcasing effectiveness under realistic and complex scenarios.

---

> ### Author Response · Authors · 2025-11-23
>
> > Response to Q1: Concerns about node-level versus edge-level explanations
>
> Node-level and edge-level explanations are both important and common in graph applications. Node-level and edge-level explanations are both important and common in graph applications. Our method is not limited to node-level explanation formats. In Section 5.4, we evaluate the method with PGExplainer, which provides edge-level explanations. For edge-level explanations, we convert edge importance scores to node importance by averaging the scores of all edges connected to each node. This conversion allows our ranking-based framework to incorporate edge-level explanations. The method remains effective with PGExplainer, demonstrating that our ranking-based approach generalizes beyond node-level scenarios to different explanation types.
>
>
>
> > Response to Q2: Concerns about robustness to noisy explanations
>
> **Motivation of experiments on noisy explanations:** Explanations can become noisy in practical scenarios where explainers have inherent uncertainty and errors. In addition, service providers may intentionally add noise as a defense mechanism to protect model intellectual property through differential privacy[1]. Evaluating robustness to noisy explanations could ensure practical viability when real-world conditions introduce perturbations or uncertainties.
>
> **Why our method is robust to explanation noise:** Our method is robust to explanation noise because our ranking-based alignment focuses on ordinal information (relative ranking) rather than absolute score values.
> - Noises on explanations usually have limited impact on the relative order of node importance. Therefore, the rankings still can partially reflect the decision logic of the target model.
> - Our data augmentation strategy further enhances robustness by creating diverse graph variations, which prevents overfitting to specific noisy patterns.
> - Additionally, even noisy explanations contain useful information about the target model's decision logic, which our framework can effectively capture and utilize.
>
> [1] Deep Learning with Differential Privacy

---

> > ### Comment · Reviewer_kv62 · 2025-11-24
> >
> > Thank you for the rebuttal. I would like to raise one central question:
> >
> > According to the paper’s claims, should the model stealing paradigm be fundamentally transformed? For example, does it imply that an attacker could (1) first train a local explainer (since methods like PGExplainer can be trained entirely offline), and (2) then use the explainer-generated explanations together with predictions to train the classifier, leading to a classifier that performs better than one trained solely on predictions?

---

> > > ### Author Response · Authors · 2025-11-25
> > > **Response to the follow-up question**
> > >
> > > Thank you for raising this thoughtful question. We would like to clarify two key points.
> > >
> > >
> > > >“According to the paper’s claims, should the model stealing paradigm be fundamentally transformed?”
> > >
> > > We’d like to further clarify the intuition behind why our EGSteal is robust to various explainers. Intuitively, although different explainers adopt different mechanisms, their explanations can partially reflect the model's decision logic from different perspectives. Hence, EGSteal adopts the rank-based explanation alignment loss to capture useful information from these explanations. In fact, we have tested different explainer types including gradient-based explanations, optimization-based post-hoc explanations, and even noisy explanations in Sec. 5.3 and Sec. 5.4.
> > >
> > > >”does it imply that an attacker could (1) first train a local explainer (since methods like PGExplainer can be trained entirely offline), and (2) then use the explainer-generated explanations together with predictions to train the classifier, leading to a classifier that performs better than one trained solely on predictions?”
> > >
> > > While conceptually possible, this approach faces significant practical challenges.
> > >
> > > Specifically, explainers generally require access to internal information of the target model, such as gradients or embeddings. For instance, PGExplainer needs gradients from the target model to optimize an MLP for important subgraph generation. However, in the model stealing scenario, the attacker typically only has black-box access to the target GNN, making the training of such explainers impractical.
> > >
> > > In contrast, our EGSteal introduces a rank-based explanation alignment loss in the surrogate model training. This ensures the learned surrogate model replicates both the predictions and underlying decision logic of the target model. Additionally, to address query budget limitations, EGSteal employs explanation-guided data augmentation, supported by both theoretical and empirical analysis.
> > >
> > > Please let us know if you have further questions, we will respond promptly.

---

> > > > ### Comment · Reviewer_kv62 · 2025-11-25
> > > >
> > > > Thank you for the clarification. Let me first apologize. After rechecking the PGExplainer paper, I realized that I did misremember the details.
> > > >
> > > > What I wanted to say is that not all explanation methods require access to the model's internal information. Some classic black-box explainers, such as LIME, Kernel SHAP, and SubgraphX (please correct me if I’m wrong), treat the model purely as a black box and only rely on querying inputs and observing outputs. Because of that, an attacker could generate these explanations locally without accessing any internal information of the target model.
> > > >
> > > > From my understanding of your paper, explanations seem to always improve model stealing performance beyond using predictions alone. If that is the case, I am wondering whether an attacker could generally follow a two-step procedure in the future: first generate explanations locally using black-box explainers, and then combine these explanations with predictions to train a stronger surrogate model.
> > > >
> > > > I am genuinely interested in the claims of your work, so I will try my best to respond quickly. Thank you again for taking the time to answer my questions.

---

> > > > > ### Author Response · Authors · 2025-11-25
> > > > > **Further response to the follow-up question**
> > > > >
> > > > > Thank you for the clarification and for taking the time to revisit the details.
> > > > >
> > > > > You are correct that some explanation methods, such as SubgraphX and GraphLIME, can operate in a black-box setting without requiring access to the model's internal information. If an attacker deploys such a black-box explainer, it could generate explanations locally. In this scenario, a framework like our EGSteal would still be necessary to effectively utilize these explanations to enhance model stealing. As shown in our analysis, a surrogate model trained with explanation alignment achieves better performance than one trained solely on predictions.
> > > > >
> > > > > We would like to further discuss the practical benefits and costs of this two-step procedure.
> > > > >
> > > > > Black-box explainers such as SubgraphX and GraphLIME typically require multiple queries per sample to generate reliable explanations [1][2]. Considering realistic query-budget constraints, we identify two distinct scenarios for this two-step procedure:
> > > > >
> > > > > 1. **Limited query budget**: The attacker faces a trade-off between (1) explaining a small number of samples with high-quality explanations, or (2) explaining more samples but with lower-quality explanations due to fewer queries per sample. In either case, the coverage or quality of explanations may not be able to effectively support model stealing.
> > > > > 2. **Sufficient query budget**: When the query budget is sufficient, the attacker can conduct additional multiple queries to obtain explanations of shadow data owned by the attacker. These obtained high-quality explanations would facilitate model stealing with the EGSteal framework. This typically corresponds to the situation that the size of shadow data is quite limited and becomes a bottleneck of model stealing. Our EGSteal could be easily adopted in this situation with the deployment of black-box explainers.
> > > > >
> > > > > Finally, we’d like to again emphasize that our work focuses on the scenario where explanations are directly provided by the service alongside predictions. In this setting, the attacker obtains both prediction and explanation in a single query. Our contribution lies in demonstrating the realistic security risk when services provide explanations directly.
> > > > >
> > > > >
> > > > > [1] GraphLIME: Local Interpretable Model Explanations for Graph Neural Networks
> > > > > [2] On Explainability of Graph Neural Networks via Subgraph Explorations

---

### Author Response · Authors · 2025-12-01
**Summary of our Rebuttal to Reviewers' questions**

Dear ACs, SACs and PCs,

We summarize the the raised questions from reviewers and our response in the following:

---

### **1st Reviewer kv62**

Reviewer kv62 explicitly expressed potential misunderstandings regarding our work, stating: “I may have misunderstood some points”, and raised the following questions:
(i) the realism and practicality of our problem setting, including whether explanation-providing services exist in IP-sensitive scenarios and whether query budget assumptions are realistic (W1, W2, W3, Q3); (ii) misunderstanding that our method requires knowledge of the target's explainer type (W5); (iii) justification of our theoretical assumptions about causal/style subgraph partitioning (W4); (iv) generalization to edge-level explanations (Q1); and (v) robustness of noisy explanation experiments (Q2).

We systematically addressed all concerns:
- **Problem setting realism**: We clarified the emerging regulatory context (EU AI Act, FDA guidelines, GDPR) that mandates explanations in high-stake domains, and provided evidence that query limits are standard in commercial API services and established model stealing research.
- **Explainer flexibility**: We clarified that our method does not assume knowledge of the target's explanation mechanism, as demonstrated in Section 5.4 across five different explainer types.
- **Theoretical validation**: We validated our causal/style partitioning through new subgraph fidelity experiments, showing causal subgraphs maintain high prediction consistency (over 82%) while style subgraphs exhibit substantially lower consistency.
- **Edge-level explanation generalization**: We demonstrated generalization to edge-level explanations using PGExplainer (Section 5.4) by converting edge importance to node importance.
- **Motivation of experiments on noisy explanations**: We explained the practical motivation that explanations can become noisy due to inherent explainer uncertainty or intentional noise addition as a defense mechanism. Our ranking-based alignment approach and data augmentation strategy together ensure robustness in such scenarios by focusing on ordinal information and preventing overfitting to noise.

In our final interaction, Reviewer kv62 acknowledged EGSteal's effectiveness in stealing model logic, stating: "From my understanding of your paper, explanations seem to always improve model stealing performance beyond using predictions alone," and expressed genuine interest: "I am genuinely interested in the claims of your work, so I will try my best to respond quickly." However, further interaction was terminated due to the technical incident at openreview.

---

### **2nd Reviewer 1dwp**

2nd Reviewer 1dwp explicitly stated: "If these two questions (Q2, Q3) are properly addressed, I would like to raise the score to at least positive." Specifically, the raised questions are:
(i) the scalability of EGSteal (W2, Q1);
(ii) quality of explanation-guided augmented data (W1);
(iii) ablation studies on explanation loss and flexibility with different explainers (Q2, Q3).

We systematically addressed all concerns:
- **Scalability**: We clarified that our evaluation includes large-scale graphs (Section 5.4) and reported EGSteal’s efficient training time (Appendix G.9).
- **Data augmentation quality**: We validated the high quality of augmented data (over 95% consistency) in our rebuttal.
- **Ablation studies**: We directed the reviewer to comprehensive ablations (Section 5.5, Appendices G.6–G.7), highlighting the critical role of our alignment loss and demonstrating flexibility across different explainer types.
This reviewer has not yet interacted with us. We expect our clarifications fully address the stated condition for a positive rating adjustment.

---

### **3rd Reviewer njmV**

This reviewer raised substantive questions about explanation faithfulness (W1-W4). We provided extensive new analyses: measuring faithfulness across five explainers (73% to 82%), systematically injecting noise (10% to 30%) showing robustness of our ranking-based mechanism, and conducting experiments on weak classifiers achieving 89% to 96% fidelity.

The reviewer expressed clear satisfaction and increased his overall rating from 4 to 6.

---

With our rebuttal, we have thoroughly addressed the main concerns raised by reviewers. We would greatly appreciate your consideration of these clarifications in the final decision.


Best regards,
The Authors

---

### Author Response · Authors · 2025-12-01
**Summary of Reviewer Engagement and Rating Improvements**

Dear ACs, SACs and PCs,

Though our work's initial ratings are 2, 4, 4, we'd like to highlight additional support from three reviewers not reflected in these initial scores.

**3rd Reviewer njmV** was satisfied with our rebuttal and **increased his rating from 4 to 6 at 11/25** that is 3 days before the accident.

**2nd Reviewer 1dwp provided a clear condition for rating increase: "I suggest the author to at least conduct ablation study on the questions 2&3... If these two questions are properly addressed I would like to raise the score to at least positive."** In our rebuttal, we clarified that the requested ablation studies were already present in our submission and supplemented detailed experimental results for the raising questions, expecting a positive rating adjustment.

Though **1st reviewer kv62** initially rated 2, he noted: **"I may have misunderstood some points, and I would appreciate further clarification from the authors during the rebuttal phase."** Throughout our discussion, reviewer kv62's stance evolved notably. In our final interaction, he acknowledged EGSteal's effectiveness in stealing model logic, stating: **"From my understanding of your paper, explanations seem to always improve model stealing performance beyond using predictions alone,"** and expressed genuine interest: **"I am genuinely interested in the claims of your work, so I will try my best to respond quickly."** However, further interaction was terminated due to the technical incident at openreview.

We believe the discussions demonstrate solid contributions to understanding security risks in explanation-enhanced ML services. We respectfully request that the final decision consider the progress reflected in these substantive exchanges, and thank the reviewers for their thoughtful engagement and the Area Chair for considering the evolution of this discussion.

Best regards,
Authors

---

### Meta-Review · Area_Chair_TReg · 2026-01-06

**Summary:**

The submission introduces a novel model stealing attack, EGSteal, which leverages explanation outputs from Graph Neural Networks (GNNs) to more effectively replicate a target model's behavior. It explores an interesting idea, but it is built upon a threat model of questionable practicality and a theoretical foundation that is overly simplistic and not well-justified. While one reviewer was satisfied with additional empirical results, the most critical reviews highlight flaws that strike at the heart of the paper's contribution and credibility. The work does not convincingly demonstrate a significant, realistic security risk nor does it provide a methodologically sound approach. The improvements made in the rebuttal are insufficient to overcome these fundamental issues. Therefore, the paper is not acceptable in its current form.

**Reviewer Concerns:**

Addressed Concerns:
*   The authors provided useful empirical analysis on explanation faithfulness and robustness to noise, which satisfied Reviewer njmV, leading them to raise their score.
*   They clarified that ablation studies existed in the appendices and pointed to experiments on larger node-classification graphs to address scalability.

Outstanding or Partially Addressed Concerns:
*   Threat Model Realism: The central objection regarding the practicality of the attack scenario remains. Citing future regulations does not validate the present-day relevance of attacking a service that provides explanations, a service that arguably would not exist in the high-stakes, IP-sensitive domains the paper describes.
*   Theoretical Soundness: The rebuttal does not fix the fundamental flaw in the causal/style partition assumption. Providing fidelity scores for augmented data does not retroactively justify the theoretically shaky premise upon which the augmentation is built. The approach remains built on an oversimplification that limits its credibility and generalizability.

**Reviewer Scores:**

*   Reviewer kv62: Initial score: 2 (Reject). While the final discussion showed a more engaged tone, the reviewer's foundational concerns about problem realism and practicality were not resolved. They would likely maintain a reject recommendation.
*   Reviewer 1dwp: Initial score: 4 (Marginally Below). Their request for ablation studies was addressed by reference, but their deeper concern about the "incorrectness" of the augmented data stemming from a flawed assumption (W1) was not solved. They might marginally increase their score but likely not above the acceptance threshold.
*   Reviewer njmV: Initial score: 4. This reviewer increased their score to 6 (Accept) based on the new empirical analysis.

---

### Decision · Program_Chairs · 2026-01-26

Reject